# BEST ARM IDENTIFICATION WITH CORRELATED SAMPLING

## ABSTRACT

Best arm identification (BAI) is an important research topic in sequential decision-making. In the fixed-confidence setting, the sample complexity, i.e., the number of samples needed to guarantee a given confidence level, serves as a fundamental metric for evaluating algorithms. Garivier & Kaufmann (2016) provided a tight characterization of this complexity as $\mathcal{H}^* \log(1/\delta)$, where $\mathcal{H}^*$ captures the problem hardness and $\delta$ is the confidence parameter. We improve this best-known bound to $\mathcal{T}^* \log(1/\delta)$ with a strictly smaller hardness parameter $\mathcal{T}^*$. Our approach is based on *correlated sampling*, which requires no assumptions on the reward function or the arm structures. A key theoretical challenge is that the resulting lower bound is defined by a non-convex optimization problem. To solve it, we propose an efficient method that decomposes the feasible region into sub-intervals and identifies local optima within each. Moreover, we propose the first **COR**related-**SA**mpling-based BAI algorithm, CORSA, and prove its asymptotic optimality. Finally, we conduct numerical experiments to evaluate the algorithm's performance.

## 1 INTRODUCTION

Best arm identification (BAI) is a fundamental and widely studied problem in machine learning, with broad applications in clinical trials (Villar et al., 2015), chemical engineering (Wang et al., 2024), and prompt optimization for large language models (Shi et al., 2024). The fixed-confidence BAI setting, which aims to identify the best arm with probability at least a given threshold while minimizing the required number of samples, is now well understood. Tight instance-dependent lower bound on sample complexity (Garivier & Kaufmann, 2016) and asymptotically optimal algorithms (Degenne et al., 2019; Wang et al., 2021) have been established. Building on this framework, a line of work focuses on reducing sample complexity further by exploiting similarity information about the reward function. The central idea is to incorporate similarity information about the reward function, enabling the agent to leverage such information to improve sampling efficiency and reduce sample complexity. Commonly studied similarity structures include linearity (Jedra & Proutiere, 2020), generalized linear models (Jun et al., 2021; Rivera & Tewari, 2024), Lipschitz continuity (Wang et al., 2021; Wanner et al., 2025), and kernelized reward functions (Zhu et al., 2021; Du et al., 2021). However, these approaches rely on strong assumptions that are often difficult to justify or verify in practice. This raises an important open question: can we improve the best achievable sample complexity without relying on strong structural assumptions about the reward function?

In this paper, we provide a positive answer to this open question by introducing a key concept called *correlated sampling* into the BAI problem. Correlated sampling is a widely used technique for variance reduction when comparing stochastic systems through simulation (Glasserman & Yao, 1992). For the BAI problem under correlated sampling, at each time step $t$, the agent selects an arm to sample and receives a random observation. Observations from different arms are correlated whenever they share the same replication index; for example, the $i$-th samples of different arms are correlated. In the fixed-confidence setting, the agent aims to exploit this correlation structure to estimate the mean performance of the arms more efficiently, identify the best arm with probability at least $1 - \delta$, and minimize the total number of samples required.

Compared to approaches that exploit similarity information in the reward function, correlated sampling relies on weaker assumptions, is straightforward to implement, and applies to nearly all BAI

problem instances—as long as correlation among observations can be introduced. We present two motivating examples that highlight the practical relevance of this setting. In queueing system optimization, a higher service rate typically reduces average sojourn time but also increases operational costs. The agent aims to identify the best service rate, such as the optimal number of servers, that balances these competing factors. Positive correlation can be naturally introduced by using the same sequence of customer arrivals time across different service rate configurations. The goal is then to develop an efficient sampling strategy to identify the best configuration as quickly as possible. In many personalized medicine problems, treatment decisions are guided by disease simulation models (Hur et al., 2004). The agent seeks to identify the most effective treatment plan by interacting with the simulation model and random treatment outcomes. By simulating the same virtual patient across different treatment plans, positive correlation is introduced among the observations, enabling counterfactual-like analysis at the individual level (Stout & Goldie, 2008).

We summarize the primary technical challenges and contributions of this work as follows:

- We introduce the concept of correlated sampling into the fixed-confidence BAI problem. This correlation breaks the independence assumption commonly used in canonical BAI analyses, rendering existing algorithms and theoretical guarantees inapplicable. Building on this, we establish the first tight, instance-dependent lower bound on sample complexity in this setting. The resulting bound takes the form of a non-convex min-max optimization problem, fundamentally different from the convex formulations that arise in canonical BAI. Furthermore, we demonstrate that incorporating positive correlation can lead to a reduction in sample complexity.

- To address the challenges posed by non-convexity, we conduct a detailed analysis of the min-max optimization problem. We establish key theoretical properties that reduce the non-convex formulation to a single-variable nonlinear optimization problem. By exploiting its structure, we show that the problem can be solved efficiently by decomposing the feasible interval into sub-intervals and locating the root within each. Moreover, we present the two-armed case, where the optimal sampling ratio admits a closed-form solution, providing clear intuition into how the introduced correlation influences the optimal sampling ratio.

- Building on these theoretical results, we propose CORSA, the first correlated sampling–based BAI algorithm. CORSA iteratively solves the nonlinear optimization problem and leverages the solution to guide the sampling process. We prove that CORSA attains the improved sample complexity bound and verify these theoretical results through numerical experiments.

Our study relates to three main strands of the existing literature:

**Best Arm Identification.** BAI is one of the most extensively studied problems in the multi-armed bandit literature (Audibert & Bubeck, 2010; Gabillon et al., 2012). Garivier & Kaufmann (2016) established a tight, instance-dependent lower bound on the sample complexity in the fixed-confidence setting and proposed an asymptotically optimal algorithm, Track-and-Stop. This framework has since inspired a substantial body of research, including the design of efficient and asymptotically optimal algorithms (Degenne et al., 2019; Wang et al., 2021), as well as extensions to broader settings such as partition identification (Juneja & Krishnasamy, 2019), multiple correct answers (Degenne & Koolen, 2019), multi-fidelity evaluations (Poiani et al., 2022; 2024a), and unknown variances (Jourdan et al., 2023). This paper aims to improve upon the best achievable sample complexity lower bound established by Garivier & Kaufmann (2016). However, since our method incorporates correlation, which violates the standard independence assumption, the existing theoretical results and algorithms are not directly applicable.

**BAI with Similarity Structure.** Another line of work explores methods to improve sample complexity by incorporating similarity structures in the reward function. Commonly studied structures include linear models (Jedra & Proutiere, 2020), generalized linear models (Jun et al., 2021; Rivera & Tewari, 2024), kernelized reward functions (Zhu et al., 2021; Du et al., 2021), Lipschitz continuity (Wang et al., 2021; Wanner et al., 2025), and unimodality (Poiani et al., 2024b). Compared to this line of work, our correlated sampling approach is fundamentally different: it does not rely on any structural assumptions about the unknown reward function. This generality makes our method applicable to a broader range of problem settings.

**Correlated Sampling.** The idea of leveraging positive correlation to reduce variance in the comparison of different stochastic systems originates in the simulation literature (Wright & Ramsay Jr, 1979; Glasserman & Yao, 1992). Prior work in this area has typically treated correlated sampling as standard variance-reduction techniques, without explicitly analyzing their implications for sample complexity (Chick & Inoue, 2001; Fu et al., 2007; Zhong & Hong, 2022). This work introduces correlated sampling into the BAI framework, thereby extending the application of this technique. More importantly, the associated theoretical results on sample complexity enrich the correlated sampling literature by providing a new perspective on the benefits of correlation in sequential decision-making problems.

## 2 PROBLEM FORMULATION AND SAMPLE COMPLEXITY

In this section, we formulate the BAI problem and review the asymptotically optimal sample complexity that can be achieved by any BAI algorithm.

Suppose there are $K$ arms, and we use $\mathcal{K} = \{1, 2, \ldots, K\}$ to index all the arms. Each arm $a \in \mathcal{K}$ is associated with a Gaussian random variable $X_a$ with an unknown mean $\mu_a$ and a known, common variance $\sigma^2$. In the case of unknown variance, one may instead use a conservative upper bound on $\sigma^2$ or adopt the framework proposed in (Jourdan et al., 2023). The agent's objective is to identify the arm with the largest mean based on noisy observations. Without loss of generality, we assume that the means are ordered in descending order throughout this paper, i.e., $\mu_1 > \mu_2 \geq \ldots \geq \mu_K$, so that arm 1 is the unique best arm. This assumption, commonly used in the BAI literature (Garivier & Kaufmann, 2016), can be relaxed to the setting where the goal is to identify an $\epsilon$-optimal arm (Degenne & Koolen, 2019).

**Learning Problem.** In the online setting, at each time step $t$, the agent selects an arm $a_t$ to sample and then observes a random outcome $Y_t$, drawn independently from the distribution of the corresponding random variable $X_{a_t}$. We denote by $\mathcal{F}_t = \sigma(a_1, Y_1, \ldots, a_t, Y_t)$ the sigma-algebra generated by the sampling decisions and observations up to time step $t$.

A BAI algorithm is characterized by three components: the sampling rule $\{a_t\}_t$, which specifies the arm to pull at time step $t$ based on the past history and is $\mathcal{F}_{t-1}$-measurable; the stopping rule $\tau$, which determines when to terminate and is a stopping time with respect to $\mathcal{F}_t$; and the decision rule $\hat{a}_\tau$, which outputs the recommended arm at termination and is $\mathcal{F}_\tau$-measurable. In the fixed-confidence setting, given a confidence level $\delta \in (0, 1)$, the agent aims to identify the best arm (arm 1) with probability at least $1 - \delta$, while minimizing the sample complexity $\mathbb{E}[\tau]$.

**Asymptotically Optimal Sample Complexity.** In this subsection, we review the best-known lower bound on the sample complexity achievable by any BAI algorithm, which serves as a benchmark for evaluating algorithmic performance. An algorithm is said to be valid if it identifies the best arm with probability at least $1 - \delta$. According to Theorem 1 of Garivier & Kaufmann (2016), the sample complexity of any valid BAI algorithm must satisfy the following lower bound:

$$\liminf_{\delta \to 0} \frac{\mathbb{E}[\tau]}{\log(1/\delta)} \geq \mathcal{H}^*(\mu) = \min_{\omega \in \Omega} \max_{a \in \mathcal{K}'} \frac{2\sigma^2 \left( \frac{1}{\omega_1} + \frac{1}{\omega_a} \right)}{(\mu_1 - \mu_a)^2}, \tag{1}$$

where $\Omega = \{\omega \in \mathbb{R}_+^K : \sum_{a \in \mathcal{K}} \omega_a = 1\}$ denotes the probability simplex, $\mathcal{K}' = \{2, \ldots, K\}$ is the index set of suboptimal arms, and $\omega_a$ represents the sampling ratio assigned to arm $a \in \mathcal{K}$.

This result implies that as the confidence level $\delta$ tends to zero, the minimal sample complexity achievable by any algorithm is $\mathcal{H}^*(\mu) \log(1/\delta)$. The quantity $\mathcal{H}^*(\mu)$ captures the instance-dependent hardness of the problem. In the Gaussian reward setting, it depends on the variance and the optimality gaps between the best arm and each suboptimal arm, while the $\log(1/\delta)$ term reflects the difficulty imposed by the confidence requirement. An algorithm is said to be asymptotically optimal if it is valid and its sample complexity satisfies

$$\limsup_{\delta \to 0} \frac{\mathbb{E}[\tau]}{\log(1/\delta)} \leq \mathcal{H}^*(\mu) = \min_{\omega \in \Omega} \max_{a \in \mathcal{K}'} \frac{2\sigma^2 \left( \frac{1}{\omega_1} + \frac{1}{\omega_a} \right)}{(\mu_1 - \mu_a)^2}. \tag{2}$$

In the BAI literature, the quantity $\mathcal{H}^*(\mu)$ serves as a key benchmark for evaluating algorithms. Several algorithms (Garivier & Kaufmann, 2016; Degenne et al., 2020; Wang et al., 2021) have

been shown to match $\mathcal{H}^*(\mu)$ asymptotically. This raises a fundamental question: can we improve upon $\mathcal{H}^*(\mu)$ without imposing restrictive assumptions on the unknown reward function? A positive answer would not only reshape existing algorithmic design but also enhance performance across a wide range of BAI problems.

## 3 IMPROVED SAMPLE COMPLEXITY

In this section, we first introduce the correlated sampling method and demonstrate its ability to improve the optimal sample complexity by comparing it with existing results. Since the improved sample complexity is characterized by a min–max optimization problem, we then analyze key theoretical properties of this problem and the corresponding optimal sampling ratios, which provide useful guidance for algorithm design. Finally, we conclude with insights from the two-armed BAI problem, which explicitly reveal the effect of correlation on the optimal sampling ratio.

### 3.1 IMPROVED SAMPLE COMPLEXITY THROUGH CORRELATED SAMPLING

Correlated sampling is a variance-reduction technique that has been extensively studied in the simulation literature (Glasserman & Yao, 1992). The key idea is that by introducing a positive correlation between two random variables, the variance of their sample mean difference is reduced, yielding a more accurate estimate of the true mean difference. However, this method has not yet been explored in the BAI literature, and the potential impact of correlation on sample complexity remains unclear.

We introduce the correlated sampling method for BAI as follows. Let $Y_a^{(l)}$ denote the $l$-th sample from arm $a$. We assume that, for a fixed replication index $l$, samples across different arms are dependent, while samples from different replications are independent. The correlation between arms $a$ and $b$ is characterized by the Pearson correlation coefficient $\rho$, defined as

$$\rho = \frac{\text{Cov}(Y_a^{(l)}, Y_b^{(l)})}{\sigma^2} > 0, \tag{3}$$

where $\text{Cov}(Y_a^{(l)}, Y_b^{(l)})$ denotes the covariance between $Y_a^{(l)}$ and $Y_b^{(l)}$. This correlation structure can be easily implemented using common random numbers across arms for the same replication index $l$, a technique widely used in the simulation literature (Fu et al., 2007). As with the variance assumption, we assume a known, common correlation coefficient $\rho$ for notational simplicity, although the framework can be readily extended to settings with heterogeneous variances and correlations. Theorem 1 presents the new sample complexity lower bound for any valid BAI algorithm under correlated sampling.

**Theorem 1.** *For any confidence level $\delta \in (0, 1)$, the sample complexity of a BAI algorithm that guarantees $\mathbb{P}(\hat{a}_\tau = 1) \geq 1 - \delta$ must satisfy*

$$\mathbb{E}[\tau] \geq \mathcal{T}^*(\mu)\, kl(\delta, 1-\delta), \qquad \liminf_{\delta \to 0} \frac{\mathbb{E}[\tau]}{\log(1/\delta)} \geq \mathcal{T}^*(\mu), \tag{4}$$

*where*

$$\mathcal{T}^*(\mu) = \min_{\omega \in \Omega} \mathcal{T}(\mu, \omega) = \begin{cases} \max_{a \in \mathcal{K}'} \dfrac{2\sigma^2 \left( \dfrac{(\rho-1)^2}{\omega_1} + \dfrac{1-\rho^2}{\omega_a} \right)}{(\mu_1 - \mu_a)^2} & \text{if} \quad \omega_1 \geq \omega_a, \\[4mm] \max_{a \in \mathcal{K}'} \dfrac{2\sigma^2 \left( \dfrac{(\rho-1)^2}{\omega_a} + \dfrac{1-\rho^2}{\omega_1} \right)}{(\mu_1 - \mu_a)^2} & \text{if} \quad \omega_a \geq \omega_1. \end{cases} \tag{5}$$

**Technical Novelty.** The analysis of Theorem 1 builds on the classical change-of-measure arguments in multi-armed bandits (Garivier & Kaufmann, 2016). However, the introduction of correlation introduces additional technical challenges. In contrast to the independent case, where the Kullback-Leibler (KL) divergence between two BAI problem instances can be decomposed as the sum of the KL divergences of individual arms, the divergence under correlated samples is more complex. We provide a detailed analysis of the KL divergence between two BAI instances, which results

in a piecewise objective function. This divergence is then related to the sample complexity and confidence level $\delta$ via the Transportation Lemma (Kaufmann et al., 2016). By considering the cases $\omega_1 \geq \omega_a$ and $\omega_a \geq \omega_1$ separately, and using the definition of the KL divergence for multidimensional Gaussian vectors, we derive the closed-form min-max optimization problems described in (5). A key distinction in Theorem 1 is that the sample complexity is characterized by a non-convex optimization problem, in contrast to the convex problem in the canonical BAI setting.

**Comparison to Existing Result.** The existing sample complexity lower bound $\mathcal{H}^*(\mu)$ arises as a special case of Theorem 1 with $\rho = 0$. In the absence of correlation across arms, we recover

$$\mathcal{T}^*(\mu) = \min_{\omega \in \Omega} \max_{a \in \mathcal{K}'} \frac{2\sigma^2 \left( \dfrac{(\rho-1)^2}{\omega_1} + \dfrac{1-\rho^2}{\omega_a} \right)}{(\mu_1 - \mu_a)^2} = \mathcal{H}^*(\mu), \tag{6}$$

which corresponds to the canonical independent case.

Moreover, we show that introducing positive correlation can strictly improve the asymptotically optimal sample complexity. In particular, we have

$$\mathcal{H}^*(\mu) > \mathcal{T}^*(\mu) = \begin{cases} \min\limits_{\omega \in \Omega} \max\limits_{a \in \mathcal{K}'} \dfrac{2\sigma^2 \left( \dfrac{1}{\omega_1} + \dfrac{1}{\omega_a} - \dfrac{2\rho}{\omega_1} + \rho^2 \left( \dfrac{1}{\omega_1} - \dfrac{1}{\omega_a} \right) \right)}{(\mu_1 - \mu_a)^2} & \text{if} \quad \omega_1 \geq \omega_a, \\[4mm] \min\limits_{\omega \in \Omega} \max\limits_{a \in \mathcal{K}'} \dfrac{2\sigma^2 \left( \dfrac{1}{\omega_1} + \dfrac{1}{\omega_a} - \dfrac{2\rho}{\omega_a} + \rho^2 \left( \dfrac{1}{\omega_a} - \dfrac{1}{\omega_1} \right) \right)}{(\mu_1 - \mu_a)^2} & \text{if} \quad \omega_a \geq \omega_1, \end{cases} \tag{7}$$

which implies that the minimal achievable sample complexity under positive correlation is $\mathcal{T}^*(\mu) \log(1/\delta)$ under positive correlation, strictly smaller than $\mathcal{H}^*(\mu) \log(1/\delta)$ in the independent case. The key intuition is that incorporating positive correlation reduces the uncertainty when comparing arms, thereby requiring fewer samples to identify the best arm.

From Theorem 1, the optimal ratio $\omega^*(\mu)$ is characterized as the solution of a non-convex min-max optimization problem. This ratio balances the variance term, the effect of positive correlation, and the optimality gaps between the best arm and the suboptimal arms. Let $N_a(t)$ denote the number of samples allocated to arm $a$ up to time step $t$, and define the corresponding sampling ratio as $\omega_a(t) = N_a(t)/t$. Intuitively, an optimal algorithm must ensure that the empirical sampling ratio $\omega(t)$ converges to the optimal ratio $\omega^*(\mu)$. Thus, designing an optimal algorithm requires solving the optimization problem and analyzing the structure of the optimal sampling ratio. For simplicity, we omit the dependence of $\omega^*(\mu)$ on $\mu$ whenever it is clear from the context.

## 3.2 OPTIMAL SAMPLING RATIO

In this subsection, we study the min-max optimization problem (5), which provides deeper insights into the optimal sampling ratio $\omega^*$ and plays a key role in designing an optimal algorithm.

The following Lemma 1 establishes a key property of the optimal sampling ratio. Intuitively, the hardness of each suboptimal arm is characterized by $F_a(\omega, \mu)$, which is proportional to its variance and inversely related to the optimality gap. The optimal sampling ratio equalizes this hardness across all suboptimal arms.

**Lemma 1.** *The optimal sampling ratio $\omega^*$ satisfies*

$$F_a(\omega^*, \mu) = F_b(\omega^*, \mu), \quad \forall a, b \in \mathcal{K}',$$

*where for any suboptimal arm $a \in \mathcal{K}'$,*

$$F_a(\omega, \mu) = \begin{cases} \dfrac{2\sigma^2 \left( \dfrac{(\rho-1)^2}{\omega_1} + \dfrac{1-\rho^2}{\omega_a} \right)}{(\mu_1 - \mu_a)^2} & \text{if} \quad \omega_1 \geq \omega_a, \\[5mm] \dfrac{2\sigma^2 \left( \dfrac{(\rho-1)^2}{\omega_a} + \dfrac{1-\rho^2}{\omega_1} \right)}{(\mu_1 - \mu_a)^2} & \text{if} \quad \omega_a \geq \omega_1. \end{cases} \tag{8}$$

To obtain the optimal sampling ratio $\omega^*$, we need to solve the non-convex optimization problem in (5). However, directly solving this problem is challenging due to its non-convexity. To address this difficulty, we leverage the key property in Lemma 1 to derive an implicit solution for $\omega^*$ in Theorem 2. The main advantage of Theorem 2 is that it reduces the original non-convex min-max optimization problem to a single-variable nonlinear optimization problem in (11), which will be the focus of our subsequent analysis.

**Theorem 2.** *The optimal sampling ratio $\omega^*$ satisfies, for any suboptimal arm $a \in \mathcal{K}'$,*

$$
\omega_a^* = \begin{cases} \dfrac{2\sigma^2(1-\rho^2)\omega_1^*}{x^*(\mu_1-\mu_a)^2 - 2\sigma^2(\rho-1)^2}, & \text{if} \quad x^*(\mu_1-\mu_a)^2 \geq 4\sigma^2(1-\rho) \\ \dfrac{2\sigma^2(\rho-1)^2\omega_1^*}{x^*(\mu_1-\mu_a)^2 - 2\sigma^2(1-\rho^2)}, & \text{if} \quad x^*(\mu_1-\mu_a)^2 < 4\sigma^2(1-\rho) \end{cases} \tag{9}
$$

*The sampling ratio for the optimal arm is*

$$
\omega_1^* = \left[ 1 + \sum_{a \in \mathcal{K}_1} \frac{2\sigma^2(1-\rho^2)}{x^*(\mu_1-\mu_a)^2 - 2\sigma^2(\rho-1)^2} + \sum_{a \in \mathcal{K}_2} \frac{2\sigma^2(\rho-1)^2}{x^*(\mu_1-\mu_a)^2 - 2\sigma^2(1-\rho^2)} \right]^{-1}. \tag{10}
$$

*Here, $x^*$ is the solution to the following single-variable nonlinear optimization problem*

$$
\min_{x \in \mathcal{F}(x)} g(x) = x + \sum_{a \in \mathcal{K}_1} \frac{2\sigma^2(1-\rho^2)x}{x(\mu_1-\mu_a)^2 - 2\sigma^2(\rho-1)^2} + \sum_{a \in \mathcal{K}_2} \frac{2\sigma^2(\rho-1)^2 x}{x(\mu_1-\mu_a)^2 - 2\sigma^2(1-\rho^2)}, \tag{11}
$$

*with $\mathcal{K}_1 = \{a \in \mathcal{K}' : x(\mu_1-\mu_a)^2 \geq 4\sigma^2(1-\rho)\}$, $\mathcal{K}_2 = \{a \in \mathcal{K}' : x(\mu_1-\mu_a)^2 < 4\sigma^2(1-\rho)\}$, and $\mathcal{F}(x) = \{x \geq 2\sigma^2(1-\rho^2)/(\mu_1-\mu_a)^2, \forall a \in \mathcal{K}_2\}$.*

The optimization problem (11) is challenging because the index sets of arms, $\mathcal{K}_1$ and $\mathcal{K}_2$, as well as the feasible region $\mathcal{F}(x)$, depend on the decision variable $x$, and the problem may possess multiple local optima. To address these difficulties, we provide a detailed analysis of its theoretical properties. The key idea is to decompose the interval $[0, +\infty)$ into several sub-intervals. Within each sub-interval, the problem can be solved efficiently, and the global optimum can then be determined by comparing the local optima across all sub-intervals.

Define $C_a = 4\sigma^2(1-\rho)/(\mu_1-\mu_a)^2$, and let $C^{(i)}$ denote the $i$-th smallest element in the set $\{C_a, a \in \mathcal{K}\}$, with $C^{(0)} = 0$ and $C^{(K)} = +\infty$. Corollary 1 divides the interval $[0, +\infty)$ into sub-intervals according to $\{C^{(i)}, i = 0, \ldots, K\}$, and, more importantly, shows that the corresponding objective function has at most one zero within each sub-interval. This property enables the design of a highly efficient algorithm for solving the optimization problem (11).

**Corollary 1.** *The optimization problem in (11) can be equivalently expressed as*

$$
\min_{i \in \{0, \ldots, K-1\}} \min_{x \in [C^{(i)}, C^{(i+1)}) \cap \mathcal{F}(x)} g_i(x), \tag{12}
$$

*where*

$$
g_i(x) = x + \sum_{a \in \mathcal{K}' \backslash \mathcal{V}_i} \frac{2\sigma^2(1-\rho^2)x}{x(\mu_1-\mu_a)^2 - 2\sigma^2(\rho-1)^2} + \sum_{a \in \mathcal{V}_i} \frac{2\sigma^2(\rho-1)^2 x}{x(\mu_1-\mu_a)^2 - 2\sigma^2(1-\rho^2)}, \tag{13}
$$

*with $\mathcal{V}_i = \{a \in \mathcal{K}' : 4\sigma^2(1-\rho)/(\mu_1-\mu_a)^2 > C^{(i)}\}$. Furthermore, the derivative $g_i'(x)$ has at most one zero within each interval $[C^{(i)}, C^{(i+1)}), i = 0, \ldots, K-1$.*

### 3.3 TWO-ARMED CASE

In this subsection, we consider the two-armed case, which admits a closed-form solution and reveals that the introduced correlation alters the optimal sampling ratio. When the variances are homogeneous across arms, the equal sampling ratio is optimal. To obtain more insightful results, we therefore focus on the heteroscedastic setting.

Consider a two-arm BAI problem instance with mean vector $\mu = (\mu_1, \mu_2)$ and known variances $\sigma_1^2$ and $\sigma_2^2$. Let $r = \sigma_2/\sigma_1$ denote the standard deviation ratio. Following the approach in Theorem 1,

we can derive that

$$\mathcal{T}^*(\mu) = \begin{cases} \min\limits_{\omega \in \Omega} \dfrac{2\left(\dfrac{\sigma_1^2(\rho r - 1)^2}{\omega_1} + \dfrac{\sigma_2^2(1 - \rho^2)}{\omega_2}\right)}{(\mu_1 - \mu_2)^2} & \text{if} \quad \omega_1 \geq \omega_2, \\[4ex] \min\limits_{\omega \in \Omega} \dfrac{2\left(\dfrac{\sigma_2^2(\rho/r - 1)^2}{\omega_2} + \dfrac{\sigma_1^2(1 - \rho^2)}{\omega_1}\right)}{(\mu_1 - \mu_2)^2} & \text{if} \quad \omega_2 \geq \omega_1. \end{cases} \tag{14}$$

Proposition 1 summarizes the optimal sampling ratio in the two-arm case. In the independent case, the optimal ratio is proportional to the standard deviations, i.e., $\omega_1^*/\omega_2^* = 1/r$. With positive correlation, however, the structure of the optimal sampling ratio changes, depending jointly on the standard deviation ratio $r$ and the correlation coefficient $\rho$.

**Proposition 1.** *The optimal sampling ratio satisfies that*

$$\omega_1^* = \begin{cases} \dfrac{(1/r)\sqrt{(\rho r - 1)^2/(1 - \rho^2)}}{1 + (1/r)\sqrt{(\rho r - 1)^2/(1 - \rho^2)}}, & \text{if} \quad r^2(1 - \rho^2) \leq (\rho r - 1)^2 \\[3ex] \dfrac{(1/r)\sqrt{(1 - \rho^2)/(\rho r - 1)^2}}{1 + (1/r)\sqrt{(1 - \rho^2)/(\rho r - 1)^2}}, & \text{if} \quad r^2(\rho/r - 1)^2 \geq (1 - \rho^2) \\[3ex] \dfrac{1}{2}, & \text{o.w.} \end{cases} \tag{15}$$

## 4 OPTIMAL ALGORITHM

In this section, we propose the first algorithm that achieves improved sample complexity through correlated sampling. We show that the proposed algorithm asymptotically attains the improved lower bound on sample complexity established in Theorem 1.

### 4.1 CORSA ALGORITHM FRAMEWORK

In this subsection, we present the general framework of the CORSA algorithm. To design an optimal algorithm, it is essential to ensure that the actual sampling ratio $\omega(t)$ converges to the optimal sampling ratio $\omega^*(\mu)$. However, there are some main challenges. First, the problem parameter $\mu$ is unknown. Second, even if the parameter $\mu$ were known, computing $\omega^*(\mu)$ requires solving a nonlinear optimization problem. Third, the stopping rule must incorporate the effect of correlation while guaranteeing both correctness and optimality. We address these challenges by carefully designing the sampling, stopping, and decision rules of the CORSA algorithm.

**Sampling Rule.** Since the problem parameter $\mu$ is unknown, a natural approach is to replace it with the empirical estimate at time step $t$:

$$\hat{\mu}_a(t) = \frac{1}{N_a(t)} \sum_{s \leq t} Y_s \mathbb{I}(a_s = a). \tag{16}$$

We then substitute the empirical parameter $\hat{\mu}(t) = \{\hat{\mu}_a(t)\}_{a \in \mathcal{K}}$ into the optimization problem (11) and solve it to obtain the empirical optimal sampling ratio $\omega^*(\hat{\mu}(t))$. Based on this, we define the following sampling rule to ensure that the actual sampling ratio $\omega(t)$ closely tracks the empirical optimal ratio:

$$a_{t+1} = \begin{cases} \arg\min_{a \in \mathcal{U}_t} N_a(t), & \text{if} \quad \mathcal{U}_t \neq \emptyset \\ \arg\max_{a \in \mathcal{K}} t\omega_a^*(\hat{\mu}(t)) - N_a(t), & \text{o.w.} \end{cases} \tag{17}$$

where $\mathcal{U}_t = \{a \in \mathcal{K} : N_a(t) \leq \sqrt{t} - K/2\}$. The sampling rule in (17) is a standard approach in the BAI literature (Garivier & Kaufmann, 2016; Juneja & Krishnasamy, 2019). Intuitively, it guarantees that each arm is sampled at least $\Omega(\sqrt{t})$ times. Asymptotically, by the law of large numbers, the empirical estimate $\hat{\mu}(t)$ converges to the true parameter $\mu$. By continuity of $\omega^*(\cdot)$, this implies that $\omega^*(\hat{\mu}(t))$ converges to $\omega^*(\mu)$.

To solve the nonlinear optimization problem (11), we first locate all zeros of the derivative of the objective function $g(x)$. Using the property in Corollary 1, the feasible interval can be decomposed into $K$ sub-regions, allowing us to compute the local optimum in each sub-region. Since each sub-region contains at most one zero, the corresponding subproblem can be solved efficiently. Finally, the results in Theorem 2 are used to determine the optimal sampling ratio.

**Stopping and Decision Rule.** The stopping rule of the algorithm is defined as follows

$$\tau = \inf\{t \in \mathbb{N} : t\mathcal{T}(\hat{\mu}(t), \omega(t))^{-1} \geq \beta(t, \delta, \rho)\}, \tag{18}$$

where the threshold function is given by

$$\beta(t, \delta, \rho) = \log\left(\frac{C(\delta, K, \rho)t^2 \log(1/\delta)^{2K+1}}{\delta}\right) \tag{19}$$

for some constant $C(\delta, K, \rho)$ that depends on the confidence level $\delta$, the number of arms $K$, and the correlation coefficient $\rho$.

Intuitively, $\beta(t, \delta, \rho)$ controls the statistical validity of the CORSA algorithm. Once the accumulated empirical evidence, measured by $t\mathcal{T}(\hat{\mu}(t), \omega(t))^{-1}$, exceeds this threshold, the algorithm stops and returns the current estimated best arm. The decision rule is then straightforward: select the arm with the largest $\hat{\mu}(\tau)$. The overall framework of the algorithm is summarized in Algorithm 1.

---

**Algorithm 1:** CORSA Algorithm

---

**Input:** Confidence level $\delta \in (0, 1)$.

1 **Initialization:** Sample each arm $n_0$ times. Update $\hat{\mu}(K)$, $\omega(n_0 K) = (1/K, \ldots, 1/K)$, and $\mathcal{U}_t$.
    Set the solution and optimal value of (11) as $x(n_0 K) = 0$, $g^*(n_0 K) = +\infty$. Set $t \leftarrow n_0 K$.

2 **while** $t\mathcal{T}(\hat{\mu}(t), \omega(t))^{-1} < \beta(t, \delta, \rho)$ **do**

3     **if** $\mathcal{U}_t \neq \emptyset$ **then**

4         $a_{t+1} = \arg\min_{a \in \mathcal{U}_t} N_a(t)$

5     **else**

6         Calculate the sequence $\{C^{(i)}, i = 0, \ldots, K-1\}$ based on $\hat{\mu}(t)$.

7         **for** $i \leftarrow 0$ **to** $K - 1$ **do**

8             Determine the minimum of $g(x)$ in the interval $[C^{(i)}, C^{(i+1)})$.

9             Define $\eta_i^+ = \lim_{x \downarrow C^{(i)}} g_i(x)$, $\eta_i^- = \lim_{x \uparrow C^{(i+1)}} g_i(x)$, and the left endpoint
            $\phi_i = \min_{\phi \in [C^{(i)}, C^{(i+1)}) \cap \mathcal{F}(x)} \phi$.

$$z^* = \begin{cases} x \text{ such that } g_i'(x) = 0 & \text{if } \eta_i^+ < 0, \text{and } \eta_i^- > 0 \\ \phi_i & \text{o.w.} \end{cases} \tag{20}$$

            **if** $g_i(z^*) < g^*(t)$ **then**

10                 $x(t) = z^*, g^*(t) = g_i(z^*)$

11         Compute the empirical optimal sampling ratio $\omega^*(\hat{\mu}(t))$ using Theorem 2.

12         $a_{t+1} = \arg\max_{a \in \mathcal{K}} t\omega_a^*(\hat{\mu}(t)) - N_a(t)$

13     Sample the arm $a_{t+1}$ once. Set $t \leftarrow t + 1$.

14     Update $\hat{\mu}(t), \omega(t), \mathcal{U}_t, x(t) = 0$, and $g^*(t) = +\infty$.

**Output:** The estimated best arm $\hat{a}_\tau$.

---

Theorem 3 establishes both the statistical validity and the asymptotic optimality of CORSA. As the confidence level $\delta \to 0$, the empirical sampling ratio converges to the optimal ratio, while the number of samples required for exploration becomes negligible, causing the upper bound to asymptotically match the lower bound almost surely and in expectation.

**Theorem 3.** *There exists a constant $C(\delta, K, \rho)$ such that, under the stopping rule (18) with the threshold function (19), CORSA algorithm guarantees that for any BAI problem instance, $\mathbb{P}(\hat{a}_\tau = a^*) \geq 1 - \delta$. Moreover, the stopping time $\tau$ satisfies the following asymptotic optimality properties:*

$$\mathbb{P}\left(\limsup_{\delta \to 0} \frac{\tau}{\log(1/\delta)} \leq \mathcal{T}^*(\mu)\right) = 1, \quad \limsup_{\delta \to 0} \frac{\mathbb{E}[\tau]}{\log(1/\delta)} \leq \mathcal{T}^*(\mu). \tag{21}$$

## 5 NUMERICAL EXPERIMENT

In this section, we validate our theoretical results by comparing CORSA with the state-of-the-art BAI algorithm Track-and-Stop (Garivier & Kaufmann, 2016).

To obtain the theoretical optimal sampling ratio, we consider a problem instance with three arms, with mean parameters $\mu = (2.0, 1.8, 1.8)$ and common variance $\sigma^2 = 1$. The probability of correct identification and sample complexity are estimated from 200 independent runs of the algorithms.

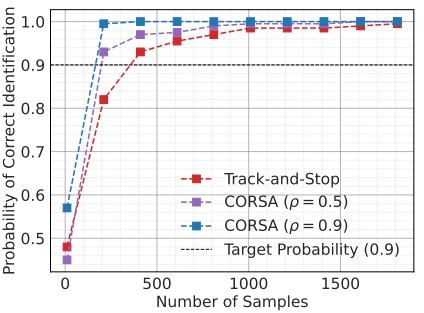

(a) Correct Probability Comparison

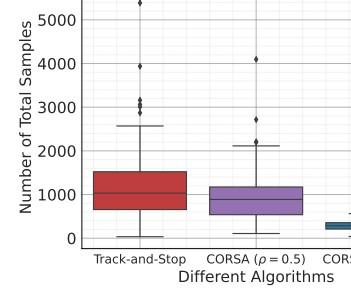

(b) Sample Complexity Comparison

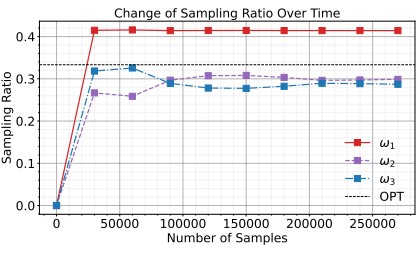

(c) Sampling Ratio of Track-and-Stop

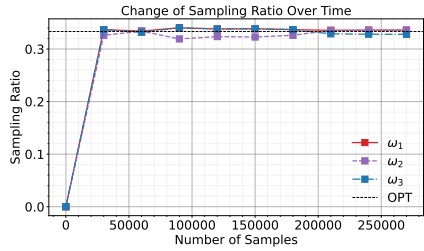

(d) Sampling Ratio of CORSA ($\rho = 0.5$)

Figure 1: Performance Comparison Between CORSA and Track-and-Stop ($\delta = 0.1$ and $n_0 = 10$)

Figure 1 compares CORSA and Track-and-Stop in terms of empirical sample complexity, probability of correct identification, and sampling ratio. The results demonstrate that CORSA outperforms Track-and-Stop in both probability of correct identification and sample complexity, highlighting that the proposed correlated sampling method improves sampling efficiency. Moreover, the gain in sample complexity becomes more pronounced as the correlation coefficient $\rho$ increases. Finally, the sampling ratio $\omega(t)$ of Track-and-Stop converges to $(0.414, 0.293, 0.293)$ with a higher complexity measure $\mathcal{H}^*(\mu) = 145.71$, whereas CORSA converges to the optimal ratio $(1/3, 1/3, 1/3)$ with a lower complexity $\mathcal{T}^*(\mu) = 75$, thereby verifying the asymptotic optimality stated in Theorem 3. The results remain consistent across different confidence levels $\delta$ (Appendix A.8).

We also conduct a queueing service rate optimization example to evaluate the practical performance of the algorithm in real-world applications. The detailed experimental setup and results are provided in Appendix A.9. The findings consistently show that CORSA is more sample-efficient than Track-and-Stop.

## 6 CONCLUSION

This paper shows how correlated sampling can improve the best-known sample complexity for BAI under the fixed-confidence setting. We establish an instance-dependent lower bound and propose CORSA, an asymptotically optimal algorithm that achieves this bound. Unlike existing methods, our approach is flexible, reward-function independent, and easy to implement. These results introduce a new algorithmic framework for BAI with correlated sampling and offer fresh insights into the role of correlation in sequential decision-making. Future directions include extensions to heterogeneous correlations and applications to best-policy identification in reinforcement learning.

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

# A APPENDIX

## A.1 LARGE LANGUAGE MODELS USAGE

ChatGPT was used for wording refinement and expression improvement.

## A.2 PROOF OF THEOREM 1.

For every $t \geq 1$, let $N_a(t)$ denote the random number of samples allocated to arm $a$ up to time step $t$. Sort the arms in descending order based on the number of samples obtained, and use the subscript $(a)$ to denote the arm ranked $a$-th in this ordering. Since, for a fixed replication index $l$, the observations are dependent, the KL divergence between two BAI problem instances $\mu$ and $\lambda$ can be expressed as

$$\sum_{a \in \mathcal{K}} \mathbb{E}[N_{(a)}(t) - N_{(a-1)}(t)] \text{KL}(\mu_{(a)}, \ldots, \mu_{(K)} || \lambda_{(a)}, \ldots, \lambda_{(K)}), \tag{22}$$

where we define $N_{(0)}(t) = 0$.

Let $a^*(\lambda)$ denote the best arm under problem instance $\lambda$, so that $a^*(\mu) = 1$. Then, for any alternative instance $\lambda$ with a unique best arm $a^*(\lambda) \neq 1$, the definition of a fixed-confidence algorithm implies that

$$\mathbb{P}_\mu(\hat{a}_\tau \neq 1) \leq \delta, \tag{23}$$

and

$$\mathbb{P}_\lambda(\hat{a}_\tau \neq 1) \geq 1 - \delta. \tag{24}$$

By the transportation lemma (Lemma 1 in Kaufmann et al. (2016)), for any alternative instance $\lambda$ with $a^*(\lambda) \neq 1$, we obtain

$$\sum_{a \in \mathcal{K}} \mathbb{E}[N_{(a)}(\tau) - N_{(a-1)}(\tau)] \mathrm{KL}(\mu_{(a)}, \dots, \mu_{(K)} || \lambda_{(a)}, \dots, \lambda_{(K)}) \geq \mathrm{kl}(\delta, 1 - \delta). \qquad (25)$$

Therefore, it holds that

$$\begin{aligned}
\mathrm{kl}(\delta, 1 - \delta) &\leq \sum_{a \in \mathcal{K}} \mathbb{E}[N_{(a)}(\tau) - N_{(a-1)}(\tau)] \mathrm{KL}(\mu_{(a)}, \dots, \mu_{(K)} || \lambda_{(a)}, \dots, \lambda_{(K)}) \\
&\leq \inf_{a^*(\lambda) \neq 1} \sum_{a \in \mathcal{K}} \mathbb{E}[N_{(a)}(\tau) - N_{(a-1)}(\tau)] \mathrm{KL}(\mu_{(a)}, \dots, \mu_{(K)} || \lambda_{(a)}, \dots, \lambda_{(K)}) \\
&\leq \sup_{\omega \in \Omega} \inf_{a^*(\lambda) \neq 1} \sum_{a \in \mathcal{K}} \mathbb{E}[N_{(a)}(\tau) - N_{(a-1)}(\tau)] \mathrm{KL}(\mu_{(a)}, \dots, \mu_{(K)} || \lambda_{(a)}, \dots, \lambda_{(K)}) \\
&\leq \mathbb{E}[\tau] \sup_{\omega \in \Omega} \inf_{a^*(\lambda) \neq 1} \sum_{a \in \mathcal{K}} (\omega_{(a)} - \omega_{(a-1)}) \mathrm{KL}(\mu_{(a)}, \dots, \mu_{(K)} || \lambda_{(a)}, \dots, \lambda_{(K)}),
\end{aligned}$$
$$(26)$$

where $\omega_{(a)} = \mathbb{E}[N_{(a)}(\tau)] / \mathbb{E}[\tau]$ denote the sampling ratio of the arm $(a) \in \mathcal{K}$.

Therefore, we conclude that

$$\begin{aligned}
\mathbb{E}[\tau] &\geq \left[ \sup_{\omega \in \Omega} \inf_{a^*(\lambda) \neq 1} \sum_{a \in \mathcal{K}} (\omega_{(a)} - \omega_{(a-1)}) \mathrm{KL}(\mu_{(a)}, \dots, \mu_{(K)} || \lambda_{(a)}, \dots, \lambda_{(K)}) \right]^{-1} \mathrm{kl}(\delta, 1 - \delta) \\
&= \mathcal{T}^*(\mu) \mathrm{kl}(\delta, 1 - \delta).
\end{aligned}$$
$$(27)$$

As $\delta \to 0$, $\mathrm{kl}(\delta, 1 - \delta) \sim \log(1/\delta)$. Hence, we obtain the following asymptotic lower bound

$$\liminf_{\delta \to 0} \frac{\mathbb{E}[\tau]}{\log(1/\delta)} \geq \mathcal{T}^*(\mu). \qquad (28)$$

We can further verify that

$$\mathcal{T}^*(\mu)^{-1}$$
$$= \sup_{\omega \in \Omega} \inf_{a^*(\lambda) \neq 1} \sum_{a \in \mathcal{K}} (\omega_{(a)} - \omega_{(a-1)}) \mathrm{KL}(\mu_{(a)}, \dots, \mu_{(K)} || \lambda_{(a)}, \dots, \lambda_{(K)})$$
$$= \begin{cases} \max_{\omega \in \Omega} \min_{a \in \mathcal{K}'} \inf_{\lambda_1 \leq \lambda_a} \omega_a \mathrm{KL}(\mu_1, \mu_a || \lambda_1, \lambda_a) + (\omega_1 - \omega_a) \mathrm{KL}(\mu_1 || \lambda_1), & \text{if} \quad \omega_1 \geq \omega_a, \\ \max_{\omega \in \Omega} \min_{a \in \mathcal{K}'} \inf_{\lambda_1 \leq \lambda_a} \omega_1 \mathrm{KL}(\mu_1, \mu_a || \lambda_1, \lambda_a) + (\omega_a - \omega_1) \mathrm{KL}(\mu_a || \lambda_a), & \text{if} \quad \omega_a \geq \omega_1. \end{cases}$$
$$(29)$$

We begin with the case $\omega_1 \geq \omega_a$ and consider the following optimization problem

$$\max_{\omega \in \Omega} \min_{a \in \mathcal{K}'} \inf_{\lambda_1 \leq \lambda_a} \omega_a \mathrm{KL}(\mu_1, \mu_a || \lambda_1, \lambda_a) + (\omega_1 - \omega_a) \mathrm{KL}(\mu_1 || \lambda_1). \qquad (30)$$

By the definition of the KL divergence between two-dimensional Gaussian distributions, we obtain

$$\mathrm{KL}(\mu_1, \mu_a || \lambda_1, \lambda_a) = \frac{1}{2\sigma^2(1 - \rho^2)} \left( (\mu_1 - \lambda_1)^2 - 2\rho(\mu_1 - \lambda_1)(\mu_a - \lambda_a) + (\mu_a - \lambda_a)^2 \right). \qquad (31)$$

Similarly, in the one-dimensional case, we obtain

$$\mathrm{KL}(\mu_1 || \lambda_1) = \frac{(\mu_1 - \lambda_1)^2}{2\sigma^2}. \qquad (32)$$

Then, the inner optimization problem in (30) can be rewritten as

$$\inf \frac{\omega_a}{2\sigma^2(1 - \rho^2)} \left( (\mu_1 - \lambda_1)^2 - 2\rho(\mu_1 - \lambda_1)(\mu_a - \lambda_a) + (\mu_a - \lambda_a)^2 \right) + \frac{\omega_1 - \omega_a}{2\sigma^2} (\mu_1 - \lambda_1)^2$$
$$\text{s.t.} \quad \lambda_1 \leq \lambda_a.$$
$$(33)$$

The Lagrangian corresponding to this optimization problem is

$$
\begin{aligned}
L(\lambda, \nu) = &\frac{\omega_a}{2\sigma^2(1-\rho^2)}\bigg( (\mu_1 - \lambda_1)^2 - 2\rho(\mu_1 - \lambda_1)(\mu_a - \lambda_a) \\
&+ (\mu_a - \lambda_a)^2 \bigg) + \frac{\omega_1 - \omega_a}{2\sigma^2}(\mu_1 - \lambda_1)^2 + \nu(\lambda_1 - \lambda_a).
\end{aligned}
\tag{34}
$$

The KKT conditions are given by

$$
\begin{aligned}
\frac{\omega_a}{\sigma^2(1-\rho^2)}\bigg( (\lambda_1 - \mu_1) + \rho(\mu_a - \lambda_a) \bigg) + \frac{\omega_1 - \omega_a}{\sigma^2}(\lambda_1 - \mu_1) + \nu &= 0 \\
\frac{\omega_a}{\sigma^2(1-\rho^2)}\bigg( \rho(\mu_1 - \lambda_1) + (\lambda_a - \mu_a) \bigg) - \nu &= 0 \\
\nu(\lambda_1 - \lambda_a) &= 0,
\end{aligned}
\tag{35}
$$

where $(\lambda, \nu)$ is the primal and dual pair.

It is straightforward to verify that $\nu^* = 0$. Consequently, by the third KKT condition, the optimal solution satisfies $\lambda_1^* = \lambda_a^*$. Combining this with the first two KKT conditions, we obtain

$$
\lambda_1^* = \lambda_a^* = \frac{\omega_1 + \rho(\omega_1 - \omega_a)}{\omega_1 + \omega_a + \rho(\omega_1 - \omega_a)}\mu_1 + \frac{\omega_a}{\omega_1 + \omega_a + \rho(\omega_1 - \omega_a)}\mu_a.
\tag{36}
$$

Then, we can obtain that

$$
\lambda_a^* - \mu_a = \frac{\omega_1 + \rho(\omega_1 - \omega_a)}{\omega_1 + \omega_a + \rho(\omega_1 - \omega_a)}(\mu_1 - \mu_a), \quad \lambda_1^* - \mu_1 = \frac{\omega_a}{\omega_1 + \omega_a + \rho(\omega_1 - \omega_a)}(\mu_a - \mu_1).
\tag{37}
$$

Using these results, we find that the optimal value satisfies

$$
\begin{aligned}
&\frac{\omega_a}{2\sigma^2(1-\rho^2)}\bigg( (\mu_1 - \lambda_1^*)^2 - 2\rho(\mu_1 - \lambda_1^*)(\mu_a - \lambda_a) + (\mu_a - \lambda_a^*)^2 \bigg) + \frac{\omega_1 - \omega_a}{2\sigma^2}(\mu_1^* - \lambda_1^*)^2 \\
&= \frac{\omega_1\omega_a(\mu_1 - \mu_a)^2}{2\sigma^2(1-\rho)(\omega_1 + \omega_a + \rho(\omega_1 - \omega_a))} \\
&= \frac{(\mu_1 - \mu_a)^2}{2\sigma^2\bigg( \dfrac{(\rho-1)^2}{\omega_1} + \dfrac{1-\rho^2}{\omega_a} \bigg)}.
\end{aligned}
\tag{38}
$$

This implies that, when $\omega_1 \geq \omega_a$, the optimization problem is equivalent to

$$
\min_{\omega \in \Omega} \max_{a \in \mathcal{K}'} \frac{2\sigma^2\bigg( \dfrac{(\rho-1)^2}{\omega_1} + \dfrac{1-\rho^2}{\omega_a} \bigg)}{(\mu_1 - \mu_a)^2}.
\tag{39}
$$

Similarly, we can follow the same method show that when $\omega_a \geq \omega_1$, the corresponding optimization problem is equivalent to

$$
\min_{\omega \in \Omega} \max_{a \in \mathcal{K}'} \frac{2\sigma^2\bigg( \dfrac{(\rho-1)^2}{\omega_a} + \dfrac{1-\rho^2}{\omega_1} \bigg)}{(\mu_1 - \mu_a)^2},
\tag{40}
$$

which concludes the proof of Theorem 1. $\qquad\square$

### A.3 PROOF OF LEMMA 1.

We prove this result by contradiction. Define

$$
G_1^* = \max_{a \in \mathcal{K}'} F_a(\omega^*, \mu),
$$

and let

$$\mathcal{S}_1 = \{a \in \mathcal{K}' : F_a(\omega^*, \mu) = G_1^*\}, \tag{41}$$

where $\mathcal{K}' = \{2, \ldots, K\}$ denotes the set of suboptimal arms. Suppose the conclusion does not hold. Then there exists a nonempty set

$$\mathcal{S}_2 = \mathcal{K}' \setminus \mathcal{S}_1 \neq \emptyset.$$

Then, we know that

$$G_2^* = \max_{a \in \mathcal{S}_2} F_a(\omega^*, \mu) < \max_{a \in \mathcal{K}'} F_a(\omega^*, \mu) = G_1^*. \tag{42}$$

Next, we construct a perturbed sampling ratio $\tilde{\omega}(\epsilon)$. Consider

$$\tilde{\omega}_a(\epsilon) = \begin{cases} \omega_1^* + \epsilon, & \text{if} \quad a = 1, \\ \omega_a^* + \epsilon, & \text{if} \quad a \in \mathcal{S}_1, \\ \omega_a^* - C_0 \epsilon, & \text{if} \quad a \in \mathcal{S}_2, \end{cases} \tag{43}$$

where $C_0 = (1 + |\mathcal{S}_1|)/|\mathcal{S}_2|$ and $\epsilon \in (0, \min_{a \in \mathcal{S}_2} \omega_a^*/C_0)$.

It is straightforward to verify that $\tilde{\omega}(\epsilon) \in \Omega$, so it is a feasible solution to the optimization problem (5).

Define

$$\tilde{G}_1(\epsilon) = \max_{a \in \mathcal{K}'} F_a(\tilde{\omega}(\epsilon), \mu), \quad \tilde{G}_2(\epsilon) = \max_{a \in \mathcal{S}_2} F_a(\tilde{\omega}(\epsilon), \mu).$$

Since $F_a(\omega, \mu)$ is a monotonically decreasing function with respect to $\omega$, we have that

$$\tilde{G}_1(\epsilon) = \max_{a \in \mathcal{K}'} F(\tilde{\omega}(\epsilon), \mu) < \max_{a \in \mathcal{K}'} F_a(\omega^*, \mu) = G_1^*. \tag{44}$$

Moreover, as $\tilde{G}_2(\epsilon)$ is continuous in $\epsilon$ and $G_2^* < G_1^*$, for sufficiently small $\epsilon$ we obtain

$$\tilde{G}_2(\epsilon) < G_1^*. \tag{45}$$

Thus, under $\tilde{\omega}(\epsilon)$ the objective value is strictly smaller, implying that $\tilde{\omega}(\epsilon)$ is a better solution than $\omega^*$. This contradicts the optimality of $\omega^*$.

Therefore, we conclude that for any suboptimal arm $a, b \in \mathcal{K}'$,

$$F_a(\omega^*, \mu) = F_b(\omega^*, \mu).$$

$\square$

### A.4 PROOF OF THEOREM 2.

By Lemma 1, the optimization problem in (5) can be reformulated as

$$\min_{\omega, \mathcal{Y}} \mathcal{Y}$$

$$\text{s.t.} \quad 2\sigma^2 \left( \frac{(\rho - 1)^2}{\omega_1} + \frac{1 - \rho^2}{\omega_a} \right) = \mathcal{Y}(\mu_1 - \mu_a)^2, \quad \forall a \in \mathcal{K}_1,$$

$$2\sigma^2 \left( \frac{(\rho - 1)^2}{\omega_a} + \frac{1 - \rho^2}{\omega_1} \right) = \mathcal{Y}(\mu_1 - \mu_a)^2, \quad \forall a \in \mathcal{K}_2, \tag{46}$$

$$\sum_{a \in \mathcal{K}} \omega_a = 1, \quad \omega_a \geq 0, \quad \mathcal{Y} \geq 0,$$

where $\mathcal{K}_1 = \{a \in \mathcal{K}' : \omega_a^* \leq \omega_1^*\}, \mathcal{K}_2 = \{a \in \mathcal{K}' : \omega_a^* > \omega_1^*\}$.

Consider first $a \in \mathcal{K}_1$, i.e., $\omega_a^* \leq \omega_1^*$. Solving the equality constraint yields

$$\omega_a^* = \frac{2\sigma^2(1 - \rho^2)\omega_1^*}{\mathcal{Y}^* \omega_1^*(\mu_1 - \mu_a)^2 - 2\sigma^2(\rho - 1)^2}, \tag{47}$$

while the condition $\omega_a^* \leq \omega_1^*$ reduces to

$$\mathcal{Y}^* \omega_1^* (\mu_1 - \mu_a)^2 \geq 4\sigma^2(1 - \rho), \tag{48}$$

while the condition $\omega_a^* \geq 0$ reduces to

$$\mathcal{Y}^* \omega_1^* (\mu_1 - \mu_a)^2 \geq 2\sigma^2(\rho - 1)^2. \tag{49}$$

Since $4\sigma^2(1 - \rho) \geq 2\sigma^2(\rho - 1)^2$, the non-negativity requirement (49) is automatically satisfied whenever (48) holds.

Now consider $a \in \mathcal{K}_2$, where $\omega_a^* > \omega_1^*$. From the equality constraint we obtain

$$\omega_a^* = \frac{2\sigma^2(\rho - 1)^2 \omega_1^*}{\mathcal{Y}^* \omega_1^* (\mu_1 - \mu_a)^2 - 2\sigma^2(1 - \rho^2)}. \tag{50}$$

The condition $\omega_a^* > \omega_1^*$ is equivalent to

$$\mathcal{Y}^* \omega_1^* (\mu_1 - \mu_a)^2 < 4\sigma^2(1 - \rho), \tag{51}$$

and the non-negativity constraint $\omega_a^* \geq 0$ requires

$$\mathcal{Y}^* \omega_1^* (\mu_1 - \mu_a)^2 \geq 2\sigma^2(1 - \rho^2). \tag{52}$$

Define $x^* = \mathcal{Y}^* \omega_1^*$, and substitute $\omega_a^*$ into the normalization constraint. This yields

$$\omega_1^* \left[ 1 + \sum_{a \in \mathcal{K}_1} \frac{2\sigma^2(1 - \rho^2)}{x^* (\mu_1 - \mu_a)^2 - 2\sigma^2(\rho - 1)^2} + \sum_{a \in \mathcal{K}_2} \frac{2\sigma^2(\rho - 1)^2}{x^* (\mu_1 - \mu_a)^2 - 2\sigma^2(1 - \rho^2)} \right] = 1. \tag{53}$$

Multiplying both sides of the equation by $\mathcal{Y}^*$, we can obtain

$$\mathcal{Y}^* = x^* + \sum_{a \in \mathcal{K}_1} \frac{2\sigma^2(1 - \rho^2) x^*}{x^* (\mu_1 - \mu_a)^2 - 2\sigma^2(\rho - 1)^2} + \sum_{a \in \mathcal{K}_2} \frac{2\sigma^2(\rho - 1)^2 x^*}{x^* (\mu_1 - \mu_a)^2 - 2\sigma^2(1 - \rho^2)}, \tag{54}$$

Hence, the original optimization problem is equivalent to

$$\min_{x \in \mathcal{F}(x)} g(x) = x + \sum_{a \in \mathcal{K}_1} \frac{2\sigma^2(1 - \rho^2) x}{x (\mu_1 - \mu_a)^2 - 2\sigma^2(\rho - 1)^2} + \sum_{a \in \mathcal{K}_2} \frac{2\sigma^2(\rho - 1)^2 x}{x (\mu_1 - \mu_a)^2 - 2\sigma^2(1 - \rho^2)}, \tag{55}$$

where the feasible region is determined by the non-negativity conditions,

$$\mathcal{F}(x) = \left\{ x \geq \frac{2\sigma^2(1 - \rho^2)}{(\mu_1 - \mu_a)^2}, \forall a \in \mathcal{K}_2 \right\}. \tag{56}$$

$\square$

## A.5 PROOF OF COROLLARY 1.

By the definition of $\{C^{(i)}, i = 0, \ldots, K - 1\}$, the interval $[0, +\infty)$ can be decomposed as

$$[0, +\infty) = \bigcup_{i=0}^{K-1} [C^{(i)}, C^{(i+1)}). \tag{57}$$

Therefore, it is straightforward to verify that the optimization problem in (11) is equivalent to

$$\min_{i \in \{0, \ldots, K-1\}} \min_{x \in [C^{(i)}, C^{(i+1)}) \cap \mathcal{F}(x)} g_i(x), \tag{58}$$

where

$$g_i(x) = x + \sum_{a \in \mathcal{K}' \setminus \mathcal{V}_i} \frac{2\sigma^2(1 - \rho^2) x}{x (\mu_1 - \mu_a)^2 - 2\sigma^2(\rho - 1)^2} + \sum_{a \in \mathcal{V}_i} \frac{2\sigma^2(\rho - 1)^2 x}{x (\mu_1 - \mu_a)^2 - 2\sigma^2(1 - \rho^2)}, \tag{59}$$

and $\mathcal{V}_i = \{a \in \mathcal{K}' : 4\sigma^2(1 - \rho)/(\mu_1 - \mu_a)^2 > C^{(i)}\}$.

Next, we show that the function $g_i(x)$ has at most one zero on the interval $[C^{(i)}, C^{(i+1)})$ for each $i \in \{0, \ldots, K - 1\}$. It is straightforward to verify that $g_i(x)$ is smooth on $[C^{(i)}, C^{(i+1)})$ for all $i \in \{0, \ldots, K - 1\}$. Hence, we have

$$
g_i'(x) = 1 - \sum_{a \in \mathcal{K}' \setminus \mathcal{V}_i} \frac{4\sigma^4(1 - \rho^2)(\rho - 1)^2}{[x(\mu_1 - \mu_a)^2 - 2\sigma^2(\rho - 1)^2]^2} - \sum_{a \in \mathcal{V}_i} \frac{4\sigma^4(1 - \rho^2)(\rho - 1)^2}{[x(\mu_1 - \mu_a)^2 - 2\sigma^2(1 - \rho^2)]^2}
$$

$$
= 1 - \sum_{a \in \mathcal{K}'} \frac{B_a}{(x - D_a)^2}, \tag{60}
$$

where

$$
B_a = \frac{4\sigma^4(1 - \rho^2)(\rho - 1)^2}{(\mu_1 - \mu_a)^4}, \quad \text{and} \quad D_a = \begin{cases} \dfrac{2\sigma^2(\rho - 1)^2}{(\mu_1 - \mu_a)^2}, & \text{if } a \in \mathcal{K} \setminus \mathcal{V}, \\ \dfrac{2\sigma^2(1 - \rho^2)}{(\mu_1 - \mu_a)^2}, & \text{if } a \in \mathcal{V}. \end{cases}
$$

Define the function $f_i(x) = g_i'(x)x^2$. Then we have

$$
f_i(x) = g_i'(x)x^2 = x^2 - \sum_{a \in \mathcal{K}} B_a + \sum_{a \in \mathcal{K}} \frac{-2B_a D_a(x - D_a/2)}{(x - D_a)^2}. \tag{61}
$$

A direct calculation gives

$$
f_i'(x) = 2x + \sum_{a \in \mathcal{K}} \frac{2B_a D_a x(x - D_a)}{(x - D_a)^4}. \tag{62}
$$

It is straightforward to verify that $B_a D_a > 0$. Since $x \in [C^{(i)}, C^{(i+1)})$, when $a \in \mathcal{V}_i$, we have $D_a = 2\sigma^2(1 - \rho^2)/(\mu_1 - \mu_a)^2$, and

$$
\mathcal{Y}(\mu_1 - \mu_a)^2 = 2\sigma^2 \left( \frac{(\rho - 1)^2}{\omega_a} + \frac{1 - \rho^2}{\omega_1} \right) > 2\sigma^2 \frac{1 - \rho^2}{\omega_1}, \tag{63}
$$

which means $x = \mathcal{Y}\omega_1 > 2\sigma^2(1 - \rho^2)/(\mu_1 - \mu_a)^2 = D_a$.

Similarly, when $a \in \mathcal{K}' \setminus \mathcal{V}_i$, we have $D_a = 2\sigma^2(\rho - 1)^2/(\mu_1 - \mu_a)^2$, and

$$
\mathcal{Y}(\mu_1 - \mu_a)^2 = 2\sigma^2 \left( \frac{(\rho - 1)^2}{\omega_1} + \frac{1 - \rho^2}{\omega_a} \right) > 2\sigma^2 \frac{(\rho - 1)^2}{\omega_1} \tag{64}
$$

which means $x = \mathcal{Y}\omega_1 > 2\sigma^2(\rho - 1)^2/(\mu_1 - \mu_a)^2 = D_a$.

Therefore, we conclude that $f_i'(x) > 0$ and $f_i(x)$ is strictly increasing on the interval $[C^{(i)}, C^{(i+1)})$. It follows that $g_i'(x)$ has at most one zero for $x > 0$ on this interval. $\qquad \square$

A.6    PROOF OF PROPOSITION 1.

The optimization problem in (14) is equivalent to

$$
\mathcal{T}^*(\mu) = \begin{cases} \displaystyle\min_{\omega \in \Omega} \frac{\sigma_1^2(\rho r - 1)^2}{\omega_1} + \frac{\sigma_2^2(1 - \rho^2)}{\omega_2} & \text{if } \omega_1 \geq \omega_2, \\ \displaystyle\min_{\omega \in \Omega} \frac{\sigma_2^2(\rho/r - 1)^2}{\omega_2} + \frac{\sigma_1^2(1 - \rho^2)}{\omega_1} & \text{if } \omega_2 \geq \omega_1. \end{cases} \tag{65}
$$

Consider the case $\omega_1 \geq \omega_2$, and define

$$
F(\omega_1) = \frac{\sigma_1^2(\rho r - 1)^2}{\omega_1} + \frac{\sigma_2^2(1 - \rho^2)}{1 - \omega_1}. \tag{66}
$$

Since $\omega_1^2 \geq (1 - \omega_1)^2$, we have

$$
\begin{aligned}
F'(\omega_1) &= \frac{\sigma_2^2(1 - \rho^2)}{(1 - \omega_1)^2} - \frac{\sigma_1^2(\rho r - 1)^2}{\omega_1^2} \\
&\geq \frac{\sigma_2^2(1 - \rho^2)}{\omega_1^2} - \frac{\sigma_1^2(\rho r - 1)^2}{\omega_1^2} \\
&= \frac{\sigma_2^2(1 - \rho^2) - \sigma_1^2(\rho r - 1)^2}{\omega_1^2}.
\end{aligned}
\tag{67}
$$

When $r^2(1 - \rho^2) \geq (\rho r - 1)^2$, we have $F'(\omega_1) \geq 0$, and the corresponding optimal solution is $\omega_1^* = 1/2$. Otherwise, solving $F'(\omega_1) = 0$ yields

$$
\omega_1^* = \frac{(1/r)\sqrt{(\rho r - 1)^2/(1 - \rho^2)}}{1 + (1/r)\sqrt{(\rho r - 1)^2/(1 - \rho^2)}}.
\tag{68}
$$

Next, consider the case $\omega_2 \geq \omega_1$, and define

$$
F(\omega_1) = \frac{\sigma_2^2(\rho/r - 1)^2}{1 - \omega_1} + \frac{\sigma_1^2(1 - \rho^2)}{\omega_1}.
\tag{69}
$$

Since we have $\omega_1^2 \leq (1 - \omega_1)^2$, it holds that

$$
\begin{aligned}
F'(\omega_1) &= \frac{\sigma_2^2(\rho/r - 1)^2}{(1 - \omega_1)^2} - \frac{\sigma_1^2(1 - \rho^2)}{\omega_1^2} \\
&\leq \frac{\sigma_2^2(\rho/r - 1)^2}{\omega_1^2} - \frac{\sigma_1^2(1 - \rho^2)}{\omega_1^2} \\
&= \frac{\sigma_2^2(\rho/r - 1)^2 - \sigma_1^2(1 - \rho^2)}{\omega_1^2}.
\end{aligned}
\tag{70}
$$

When $r^2(\rho/r - 1)^2 \leq (1 - \rho^2)$, we have $F'(\omega_1) \leq 0$, and the corresponding optimal solution is $\omega_1^* = 1/2$. Otherwise, let $F'(\omega_1) = 0$, we can obtain that

$$
\omega_1^* = \frac{(1/r)\sqrt{(1 - \rho^2)/(\rho r - 1)^2}}{1 + (1/r)\sqrt{(1 - \rho^2)/(\rho r - 1)^2}},
\tag{71}
$$

which completes the proof. $\qquad\square$

### A.7 PROOF OF THEOREM 3

The proof of Theorem 3 builds on several supporting lemmas. Lemma 2 establishes the statistical validity of the CORSA algorithm, while Lemma 3 proves a key continuity property. Lemma 4 recalls known results for the sampling rule. Finally, these results are combined to derive both almost-sure and expected upper bounds on the stopping time $\tau$.

**Lemma 2.** *The CORSA algorithm satisfies that* $\mathbb{P}(\hat{a}_\tau \neq 1) \leq \delta$.

*Proof.* We begin by presenting a useful property of the KL divergence for a $K$-dimensional Gaussian distribution with positive correlation $\rho > 0$.

By definition, the KL divergence between $\mathcal{N}(\mu, \Sigma)$ and $\mathcal{N}(\lambda, \Sigma)$ is

$$
KL(\mu \| \lambda) = \frac{1}{2}(\mu - \lambda)^\top \Sigma^{-1}(\mu - \lambda),
\tag{72}
$$

where the covariance matrix is

$$
\Sigma = \sigma^2((1 - \rho)I + \rho \mathbf{1}\mathbf{1}^\top),
$$

with $I$ denoting the identity matrix and $\mathbf{1}$ the all-ones vector.

Applying the Sherman–Morrison formula, we obtain

$$\Sigma^{-1} = \frac{1}{\sigma^2} \left( \frac{1}{1-\rho} I - \frac{\rho}{(1-\rho)(1+(K-1)\rho)} \mathbf{1}\mathbf{1}^\top \right). \tag{73}$$

For $K > 1$ and $\rho \in (0,1)$, we have $\Sigma^{-1} \preceq \frac{1}{(1-\rho)\sigma^2} I$, which implies that

$$\begin{aligned}
KL(\mu||\lambda) &= \frac{1}{2}(\mu-\lambda)^\top \Sigma^{-1}(\mu-\lambda) \\
&\leq \frac{1}{2(1-\rho)\sigma^2}(\mu-\lambda)^\top(\mu-\lambda) \\
&= \frac{1}{2(1-\rho)\sigma^2} \sum_{a \in \mathcal{K}} (\mu_a - \lambda_a)^2 \\
&= \frac{1}{(1-\rho)} \sum_{a \in \mathcal{K}} KL(\mu_a||\lambda_a).
\end{aligned} \tag{74}$$

This property yields an upper bound on $KL(\mu||\lambda)$ in terms of the KL divergence of the marginal distribution, thereby simplifying the analysis of the statistical validity of the CORSA algorithm.

The stopping rule of the CORSA algorithm is defined as follows

$$\tau = \inf\{t \in \mathbb{N} : t\mathcal{T}(\hat{\mu}(t), \omega(t))^{-1} \geq \beta(t, \delta, \rho)\}. \tag{75}$$

To establish the statistical validity of the CORSA algorithm, it suffices to show that

$$\mathbb{P}\left(\tau < \infty, \hat{a}_\tau \neq 1\right) \leq \delta. \tag{76}$$

We begin by noting that

$$\begin{aligned}
&\mathbb{P}\left(\tau < \infty, \hat{a}_\tau \neq 1\right) \\
\leq &\mathbb{P}\left(\exists t \in \mathbb{N}, \hat{a}_t \neq 1, t\mathcal{T}(\hat{\mu}(t), \omega(t))^{-1} \geq \beta(t, \delta, \rho)\right) \\
= &\mathbb{P}\left(\exists t \in \mathbb{N}, \hat{a}_t \neq 1, \inf_{a^*(\lambda) \neq \hat{a}_t} \sum_{a \in \mathcal{K}} [N_{(a)}(t) - N_{(a-1)}(t)] KL(\hat{\mu}_{(a)}(t), \ldots, \hat{\mu}_{(K)}(t)||\lambda_{(a)}, \ldots, \lambda_{(K)}) \geq \beta(t, \delta, \rho)\right) \\
= &\mathbb{P}\left(\exists t \in \mathbb{N}, \sum_{a \in \mathcal{K}} [N_{(a)}(t) - N_{(a-1)}(t)] KL(\hat{\mu}_{(a)}(t), \ldots, \hat{\mu}_{(K)}(t)||\mu_{(a)}, \ldots, \mu_{(K)}) \geq \beta(t, \delta, \rho)\right) \\
\leq &\mathbb{P}\left(\exists t \in \mathbb{N}, \frac{1}{1-\rho} \sum_{a \in \mathcal{K}} N_a(t) KL(\hat{\mu}_a(t)||\mu_a) \geq \beta(t, \delta, \rho)\right) \\
\leq &\sum_{t=1}^\infty \mathbb{P}\left(\sum_{a \in \mathcal{K}} N_a(t) KL(\hat{\mu}_a(t)||\mu_a) \geq (1-\rho)\beta(t, \delta, \rho)\right) \\
\leq &\sum_{t=1}^\infty e^{K+1} \left(\frac{(1-\rho)^2 \beta(t, \delta, \rho)^2 \log(t)}{K}\right)^K e^{-(1-\rho)\beta(t, \delta, \rho)}
\end{aligned} \tag{77}$$

By setting

$$\beta(t, \delta, \rho) = \log\left(\frac{Ct^2 \log(1/\delta)^{2K+1}}{\delta}\right),$$

and choosing the constant $C$, which depends on $K, \delta$, and $\rho$, sufficiently large such that

$$\sum_{t=1}^\infty e^{K+1} \left(\frac{(1-\rho)^2 \beta(t, \delta, \rho)^2 \log(t)}{K}\right)^K e^{-(1-\rho)\beta(t, \delta, \rho)} \leq \delta, \tag{78}$$

we can conclude that

$$\mathbb{P}\left(\tau < \infty, \hat{a}_\tau \neq 1\right) \leq \delta. \tag{79}$$

$\square$

**Lemma 3.** *The function $\mathcal{T}(\mu, \omega)^{-1}$ is continuous in both $\mu$ and $\omega$. Moreover, the optimal sampling ratio $\omega^*(\mu)$ satisfies $\omega_a^*(\mu) > 0$ for all $a \in \mathcal{K}$.*

*Proof.* Proof of Lemma 3 For each suboptimal arm $a \in \mathcal{K}$, define the function

$$f_a(\omega, \mu) = \begin{cases} \dfrac{(\mu_1 - \mu_a)^2}{2\sigma^2 \left( \dfrac{(\rho-1)^2}{\omega_1} + \dfrac{1-\rho^2}{\omega_a} \right)} & \text{if} \quad \omega_1 \geq \omega_a, \\[4mm] \dfrac{(\mu_1 - \mu_a)^2}{2\sigma^2 \left( \dfrac{(\rho-1)^2}{\omega_a} + \dfrac{1-\rho^2}{\omega_1} \right)} & \text{if} \quad \omega_a \geq \omega_1. \end{cases} \tag{80}$$

Then we have $\mathcal{T}(\omega, \mu)^{-1} = \min_{a \in \mathcal{K}'} f_a(\omega, \mu)$. For the cases $\omega_1 > \omega_a$ and $\omega_1 < \omega_a$, the function $f_a(\omega, \mu)$ is continuous in both $\omega$ and $\mu$.

When $\omega_1 = \omega_a$, we have

$$f_a(\omega, \mu) = \frac{(\mu_1 - \mu_a)^2}{2\sigma^2 \left( \dfrac{(\rho-1)^2}{\omega_1} + \dfrac{1-\rho^2}{\omega_a} \right)}, \tag{81}$$

which shows that $f_a(\omega, \mu)$ is also continuous in $\omega$ at the boundary. Therefore, $f_a(\omega, \mu)$ is continuous in both $\omega$ and $\mu$.

Define the set $\mathcal{B} = \{\lambda \in \mathbb{R}^K : \lambda_1 > \lambda_2 \geq, \ldots, \lambda_K\}$. By the definition of $\mu$, we have $\mu \in \mathcal{B}$. Then, there exists a sufficiently small constant $\epsilon_1 > 0$ such that, if $\|\mu - \lambda\|_\infty < \epsilon_1$, it follows that $\lambda \in \mathcal{B}$.

Then, for any $\xi > 0$, there exists a constant $0 < \epsilon_2 \leq \epsilon_1$ such that

$$|\mathcal{T}(\omega, \mu)^{-1} - \mathcal{T}(\omega, \lambda)^{-1}| = |\min_{a \in \mathcal{K}'} f_a(\omega, \mu) - \min_{a \in \mathcal{K}'} f_a(\omega, \lambda)| \leq \epsilon_2 |\min_{a \in \mathcal{K}'} f_a(\omega, \mu)| = \epsilon_2 |\mathcal{T}(\omega, \mu)^{-1}|, \tag{82}$$

which establishes the continuity of $\mathcal{T}(\omega, \mu)^{-1}$ with respect to $\mu$. The continuity of $\omega$ follows by the same argument.

Now, we show that the optimal sampling ratio $\omega^*(\mu)$ satisfies $\omega_a^*(\mu) > 0$ for all $a \in \mathcal{K}$. Suppose, for the sake of contradiction, that there exists an arm $a \in \mathcal{K}$ such that $\omega_a^*(\mu) = 0$. In this case, the corresponding $f_a(\omega, \mu) = 0$, which implies $\mathcal{T}(\omega, \mu)^{-1} = 0$. This contradicts the optimality of $\omega^*(\mu)$, since we can always select a feasible uniform sampling rule $\tilde{\omega} \in \Omega$ with $\tilde{\omega}_a = 1/K, \forall a \in \mathcal{K}$, yielding $\mathcal{T}(\omega, \mu)^{-1} > 0$. Therefore, it must hold that $\omega^*(\mu)$ satisfies $\omega_a^*(\mu) > 0$ for all $a \in \mathcal{K}$. $\square$

**Lemma 4.** *(Lemma 17 in Garivier & Kaufmann (2016)). Consider the following sampling rule*

$$a_{t+1} = \begin{cases} \arg\min_{a \in \mathcal{U}_t} N_a(t), & \text{if} \quad \mathcal{U}_t \neq \emptyset \\ \arg\max_{a \in \mathcal{K}} t\omega_a^*(\hat{\mu}(t)) - N_a(t), & \text{o.w.} \end{cases} \tag{83}$$

*where $\mathcal{U}_t = \{a \in \mathcal{K} : N_a(t) \leq \sqrt{t} - K/2\}$. Then, for every arm $a \in \mathcal{K}$, we have $N_a(t) \geq (\sqrt{t} - K/2)_+ - 1$. Furthermore, for any $\epsilon > 0$ and $t_0 > 0$ such that*

$$\sup_{t \geq t_0} \max_{a \in \mathcal{K}} \left| \omega_a^*(\hat{\mu}(t)) - \omega_a^*(\mu) \right| \leq \epsilon, \tag{84}$$

*there exists $t_1 > 0$ such that*

$$\sup_{t \geq t_1} \max_{a \in \mathcal{K}} \left| \frac{N_a(t)}{t} - \omega_a^*(\mu) \right| \leq 3(K-1)\epsilon. \tag{85}$$

We now proceed to establish the sample complexity upper bound stated in Theorem 3.

Consider the following clean event

$$\mathcal{E} = \left\{ \hat{\mu}(t) \to \mu, \max_{a \in \mathcal{K}} \left| \frac{N_a(t)}{t} - \omega_a^*(\mu) \right| \to 0 \right\}. \tag{86}$$

According to Lemma 4, the sampling rule guarantees that $N_a(t) \geq (\sqrt{t} - K/2)_+ - 1$. Applying the law of large numbers, it then follows that $\hat{\mu}(t) \to \mu$ almost surely. According to Corollary 1, the function $g_i'(x)$ has at most one zero on the interval $[C^{(i)}, C^{(i+1)})$ for each $i = 0, \ldots, K - 1$. If no zero exists on the interval $[C^{(i)}, C^{(i+1)})$, then the minimum of $g_i(x)$ occurs at one of the endpoints, $C^{(i)}$ or $C^{(i+1)}$. The search mechanism of CORSA guarantees finding the global minimum of $g(x)$, from which we can further obtain the empirical sampling ratio $\omega^*(\hat{\mu}(t))$. Since $\omega^*(\mu)$ is continuous with respect to $\mu$, then for any $\epsilon > 0$, there exists $t_0 > 0$ such that

$$\sup_{t \geq t_0} \max_{a \in \mathcal{K}} \left| \omega_a^*(\hat{\mu}(t)) - \omega_a^*(\mu) \right| \leq \frac{\epsilon}{3(K-1)}. \tag{87}$$

Applying Lemma 4, there exists some $t_1 > 0$ such that

$$\sup_{t \geq t_1} \max_{a \in \mathcal{K}} \left| \frac{N_a(t)}{t} - \omega_a^*(\mu) \right| \leq \epsilon. \tag{88}$$

Therefore, we conclude that $\mathbb{P}(\mathcal{E}) = 1$. Condition on the event $\mathcal{E}$, and by Lemma 3, $\mathcal{T}(\omega, \mu)^{-1}$ is continuous in both $\omega$ and $\mu$. Hence, for any $\epsilon > 0$, there exists $t_2 > 0$ such that, for all $t \geq t_2$, we have

$$\mathcal{T}(\omega(t), \hat{\mu}(t))^{-1} \geq (1 - \epsilon)\mathcal{T}(\omega^*(\mu), \mu)^{-1}. \tag{89}$$

Since the threshold function satisfies $\beta(t, \delta, \rho) = o(t)$, there exists $t_3 > 0$ such that, for all $t \geq t_3$, we have

$$\beta(t, \delta, \rho) \leq \log(1/\delta) + \epsilon \mathcal{T}(\omega^*(\mu), \mu)^{-1} t. \tag{90}$$

Then, the stopping time $\tau$ satisfies that

$$\begin{aligned}
\tau &= \inf\{t \in \mathbb{N} : t\mathcal{T}(\hat{\mu}(t), \omega(t))^{-1} \geq \beta(t, \delta, \rho)\} \\
&\leq t_0 + t_1 + t_2 + \inf\{t \in \mathbb{N} : t\mathcal{T}(\hat{\mu}(t), \omega(t))^{-1} \geq \log(1/\delta) + \epsilon\mathcal{T}(\omega^*(\mu), \mu)^{-1}t\} \\
&\leq t_0 + t_1 + t_2 + \inf\{t \in \mathbb{N} : t(1 - \epsilon)\mathcal{T}(\omega^*(\mu), \mu)^{-1} \geq \log(1/\delta) + \epsilon\mathcal{T}(\omega^*(\mu), \mu)^{-1}t\} \\
&= t_0 + t_1 + t_2 + \inf\{t \in \mathbb{N} : t(1 - 2\epsilon)\mathcal{T}(\omega^*(\mu), \mu)^{-1} \geq \log(1/\delta)\} \\
&= t_0 + t_1 + t_2 + \frac{\mathcal{T}(\omega^*(\mu), \mu)\log(1/\delta)}{1 - 2\epsilon}.
\end{aligned} \tag{91}$$

Therefore, we have

$$\limsup_{\delta \to 0} \frac{\tau}{\log(1/\delta)} \leq \frac{\mathcal{T}(\omega^*(\mu), \mu)}{1 - 2\epsilon}, \tag{92}$$

almost surely. By letting $\epsilon \to 0$, it follows that

$$\mathbb{P}\left( \limsup_{\delta \to 0} \frac{\tau}{\log(1/\delta)} \leq \mathcal{T}^*(\mu) \right) = 1. \tag{93}$$

Next, we derive an upper bound on the expected stopping time $\mathbb{E}[\tau]$. According to Lemma 3, $\mathcal{T}(\omega, \mu)^{-1}$ is continuous in both $\omega$ and $\mu$. Then, for any $\epsilon > 0$, there exists $\xi_1(\epsilon) > 0$ such that for any $\omega(t)$ and $\hat{\mu}(t)$ satisfy $\|\hat{\mu}(t) - \mu\|_\infty \leq \xi_1(\epsilon)$, and $\|\omega(t) - \omega^*(\mu)\|_\infty \leq \xi_1(\epsilon)$, we have

$$\mathcal{T}(\omega(t), \hat{\mu}(t))^{-1} \geq (1 - \epsilon)\mathcal{T}(\omega^*, \mu)^{-1}. \tag{94}$$

Since $\omega^*(\hat{\mu}(t)) \to \omega^*(\mu)$ almost surely, then there exists $\xi_2(\epsilon)$ such that for any $\hat{\mu}(t)$ satisfies $\|\hat{\mu}(t) - \mu\|_\infty \leq \xi_2(\epsilon)$, we have $\|\omega^*(\hat{\mu}(t)) - \omega^*(\mu)\|_\infty \leq \xi_1(\epsilon)/3(K-1)$. Let $\xi(\epsilon) = \min\{\xi_1(\epsilon), \xi_2(\epsilon)\}$ and define the event

$$\mathcal{E}_T = \bigcap_{t = T^{1/4}}^{T} \{\|\hat{\mu}(t) - \mu\|_\infty \leq \xi(\epsilon)\}. \tag{95}$$

Let $\epsilon_1 = \xi_1(\epsilon)/3(K-1)$, according to Lemma 4, we know that there exists a constant $T(\epsilon_1)$, such that for all $T > T(\epsilon_1)$, condition on the event $\mathcal{E}_T$, we have

$$\sup_{t \geq T^{1/2}} \max_{a \in \mathcal{K}} \left| \frac{N_a(t)}{t} - \omega_a^*(\mu) \right| \leq \xi_1(\epsilon), \tag{96}$$

which further implies that $\mathcal{T}(\omega(t), \hat{\mu}(t))^{-1} \geq (1 - \epsilon)\mathcal{T}(\omega^*, \mu)^{-1}$.

Then, we have

$$
\begin{aligned}
\min(\tau, T) &\leq T^{1/2} + \sum_{t=T^{1/2}}^{T} \mathbb{I}(\tau > t) \\
&\leq T^{1/2} + \sum_{t=T^{1/2}}^{T} \mathbb{I}\left(t\mathcal{T}(\omega(t), \hat{\mu}(t))^{-1} \leq \beta(t, \delta, \rho)\right) \\
&\leq T^{1/2} + \sum_{t=T^{1/2}}^{T} \mathbb{I}\left(t \leq \frac{\beta(t, \delta, \rho)\mathcal{T}(\omega^*(\mu), \mu)}{1 - \epsilon}\right) \\
&\leq T^{1/2} + \frac{\beta(t, \delta, \rho)\mathcal{T}(\omega^*(\mu), \mu)}{1 - \epsilon}.
\end{aligned}
\tag{97}
$$

Define $T_1^*(\delta) = \inf\left\{t \in \mathbb{N} : T^{1/2} + \frac{\beta(t,\delta,\rho)\mathcal{T}(\omega^*(\mu),\mu)}{1-\epsilon} < T\right\}$, then, for any $T > \max(T(\epsilon_1), T_1^*(\delta))$, we have $\mathcal{E}_T \subset (\tau \leq T)$. Therefore, we have

$$
\mathbb{E}[\tau] = \sum_{T=1}^{\infty} \mathbb{P}(\tau \geq T) \leq T(\epsilon_1) + T_1^*(\delta) + \sum_{T=T(\epsilon_1)+T_1^*(\delta)}^{\infty} \mathbb{P}(\tau \geq T) \leq T(\epsilon_1) + T_1^*(\delta) + \sum_{T=1}^{\infty} \mathbb{P}(\mathcal{E}_T^c).
\tag{98}
$$

According to Lemmas 18 and 19 of Garivier & Kaufmann (2016), we have that

$$
T(\epsilon_1) = \frac{\mathcal{T}^*(\omega^*(\mu), \mu)}{1 - \epsilon}\left(\mathcal{O}(\log(1/\delta)) + \mathcal{O}(\log\log(1/\delta))\right),
\tag{99}
$$

and $\sum_{T=1}^{\infty} \mathbb{P}(\mathcal{E}_T^c) < \infty$. Therefore,

$$
\limsup_{\delta \to 0} \frac{\mathbb{E}[\tau]}{\log(1/\delta)} \leq \frac{1}{1 - \epsilon}\mathcal{T}^*(\omega^*(\mu), \mu).
\tag{100}
$$

Letting $\epsilon \to 0$ completes the proof.

## A.8 DETAILS OF NUMERICAL EXPERIMENTS

For Track-and-Stop, we adopt the heuristic threshold $\beta(t, \delta) = \log((\log(t) + 1)/\delta)$ proposed in Garivier & Kaufmann (2016) and also used in (Wang et al., 2021). For CORSA, we use $\beta(t, \delta) = \log((\log(t) + 1)/((1 - \rho)^3\delta))$, which, although not theoretically justified, remains conservative enough to ensure correct identification.

Figure 2 compares the sample complexity of CORSA and Track-and-Stop under different confidence levels $\delta$. The results show that as $\delta$ decreases, the required number of samples increases. Moreover, CORSA consistently requires fewer samples than Track-and-Stop.

## A.9 APPLICATION EXAMPLE: QUEUEING SERVICE RATE OPTIMIZATION

In this subsection, we evaluate the performance of CORSA through a queueing service rate optimization example. Queueing service rate optimization is a fundamental problem in simulation, as it plays a key role in improving the performance and efficiency of stochastic systems (Weber & Stidham Jr, 1987).

Consider a single-server queueing system where customers arrive according to a Poisson process with rate $\lambda$, and service times are independent and identically distributed (i.i.d.) according to an exponential distribution with rate $\mu$. The performance measure of the system is defined as

$$
f(\mu) = \mathbb{E}[S(\mu)] + C\mu,
\tag{101}
$$

where $\mathbb{E}[S(\mu)]$ denotes the average sojourn time under service rate $\mu$, and $C\mu > 0$ represents the corresponding operational cost. Intuitively, increasing the service rate reduces the average sojourn

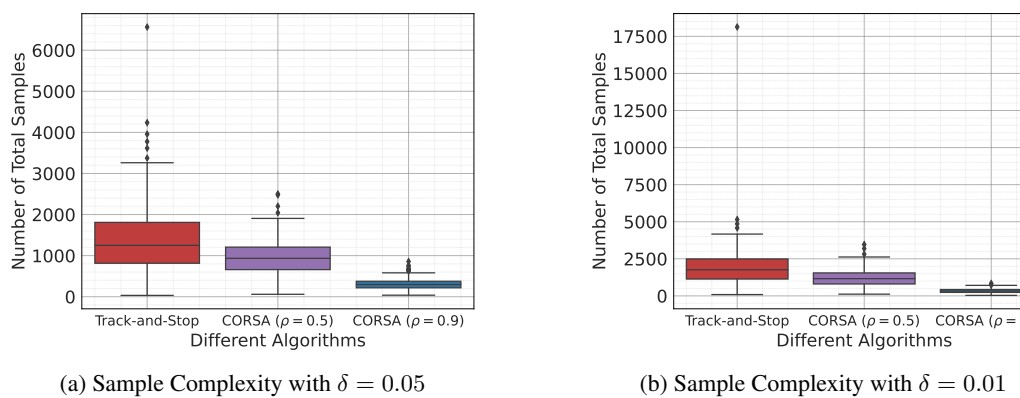

(a) Sample Complexity with $\delta = 0.05$        (b) Sample Complexity with $\delta = 0.01$

Figure 2: Performance Comparison Between CORSA and Track-and-Stop

time but also raises the operational cost. Hence, the problem requires a trade-off between system efficiency and cost.

In this example, we set the arrival rate to $\lambda = 0.5$, the unit cost to $C = 10$, and consider alternative service rates $\mu \in \{0.51, 0.53, 0.55, 0.57, 0.59\}$. At each time step $t$, the agent selects a service rate $\mu_t$ and runs a simulation model to obtain a random observation of the average sojourn time. For each simulation experiment, the average sojourn time is estimated using 40 customers. Positive correlation is naturally introduced by employing the same sequence of customer arrivals and service times across different service rate configurations.

The probability of correct identification and the sample complexity are estimated based on 200 independent runs of the algorithms. Since the variance and correlation coefficient are unknown, we adopt a homogeneous upper bound with $\sigma^2 = 20$ and $\rho = 0.5$ in the implementation.

Figure 3 compares CORSA and Track-and-Stop in terms of the empirical probability of correct identification and sample complexity. The results demonstrate that CORSA outperforms Track-and-Stop on both metrics.

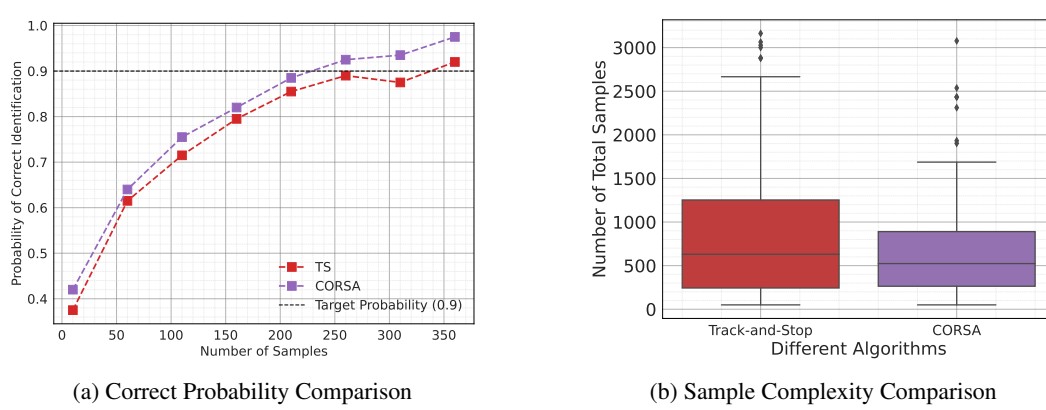

(a) Correct Probability Comparison        (b) Sample Complexity Comparison

Figure 3: Performance Comparison Between CORSA and Track-and-Stop ($\delta = 0.1, n_0 = 10$)

