# OpenReview forum: "Best Arm Identification with Correlated Sampling"
_ICLR.cc/2026/Conference — ICLR 2026 Conference Withdrawn Submission_

### Official Review · Reviewer_Afbt · 2025-10-26

**Soundness:** 3
**Presentation:** 3
**Contribution:** 3
**Rating:** 6
**Confidence:** 4

**Summary:**

This paper studies the best arm identification problem under the fixed confidence setup. It assumes the arm rewards are correlated, characterized by the correlation coefficient $\rho$. By making using of this correlation, it proposes a new algorithm CORSA, which improves the sample complexity given by Track-and-Stop, and it is proved to be asymptotically optimal. Empirical studies are carried out to demonstrate its superiority of the proposed algorithm.

**Strengths:**

- The problem setup is interesting and well motivated, as it relaxes the classical independence assumption between arm observations. The proposed algorithm is novel, theoretically sound, and proved to be asymptotically optimal. It also demonstrates superior empirical performance compared to Track-and-Stop.
- The technical challenge arising from the non-convex optimization is effectively addressed by decomposing the feasible region into intervals, enabling efficient solution of the subproblems.
- Experiments are conducted to illustrate the effectiveness of the proposed method. Although the theoretical guarantees hold in the asymptotic regime, the algorithm also performs competitively in the finite-sample setting.

**Weaknesses:**

- The current algorithm design assumes that the correlation coefficient $\rho$ is known and identical across all arm pairs, which limits its applicability in more general or heterogeneous correlation settings.
- The analysis is restricted to Gaussian bandits. Since Track-and-Stop handles single-parameter exponential family distributions, it would be valuable to extend the proposed framework to such cases.
- Experiments are limited to a 3-arm setting. Including results with a larger number of arms (e.g., 10) would strengthen the empirical evidence, especially regarding the scalability of the arm allocation solver.
- Some related works are not discussed. The paper should elaborate on the similarities and distinctions with [1, 2] to better situate its contribution within the existing literature.

[1] El Mehdi Saad and Gilles Blanchard and Nicolas Verzelen. Covariance-adaptive best arm identification. Thirty-seventh Conference on Neural Information Processing Systems. 2023.

[2] Gupta, Samarth and Joshi, Gauri and Yagan, Osman. Best-Arm Identification in Correlated Multi-Armed Bandits. IEEE Journal on Selected Areas in Information Theory. 2021.

**Questions:**

- If only an upper (or lower) bound on $\rho$ is known, how would this affect the resulting sample complexity? Furthermore, since $\rho$ is often unknown in practice, how robust is the proposed algorithm to mild misspecification of $\rho$?

- Is the current algorithm extendable to the heterogeneous-$\rho$ setting? Given that the sup-inf problem associated with the lower bound is already nontrivial to solve, can the optimization remain efficient when the correlations differ across arm pairs?

- The statement in Line 69 — “…rendering existing algorithms and theoretical guarantees inapplicable.” — seems somewhat too strong. Would it be more accurate to say that existing algorithms remain applicable but may exhibit suboptimal sample complexity under correlation?

- Could the authors clarify how the reasoning in Line 626 implies the result in Line 629? The connection between these two steps is currently unclear.

---

### Official Review · Reviewer_mZ5c · 2025-10-27

**Soundness:** 3
**Presentation:** 2
**Contribution:** 2
**Rating:** 2
**Confidence:** 4

**Summary:**

This paper investigates fixed-confidence BAI for Gaussian bandits. Distinct from canonical BAI, a bespoke correlation structure is investigated, where the observations across arms having the same replication index is equi-correlated with a known coefficient $\rho$. An instance-dependent lower bound is derived, and an algorithm named CORSA is proposed and claimed to be asymptotically optimal.

**Strengths:**

1. Under the correlation setting considered, the paper derives a sharper lower bound compared to the extensively studied fixed-confidence lower bound by Garivier and Kaufmann, and the canonical lower bound can be recovered by setting $\rho=0$.

2. A correlation-aware algorithm (CORSA) is proposed and analyzed, and shown to admit asymptotic optimality based on the derived lower bound. The key distinction from C-tracking is the solution to a non-linear optimization problem which yields the optimal arm selection based on the current estimates.

**Weaknesses:**

1. **The correlation structure:** The correlation structure, assuming that the observations sharing the same replication index should be equicorrelated, seems somewhat artificial and contrived. The authors present two motivating examples—one of a queueing system, and the other in personalised medicine—which are not very convincing. In both examples, I can agree that the observations are correlated, but I don't see how the bespoke correlation structure assumed is directly applicable. For instance, in personalised medicine, it is far more natural to construct a causal graph and treat the problem as a causal bandit as opposed to the specific correlation assumed in the paper.

2. **overclaims:** The abstract mentions: "no assumptions on the reward function or the arm structures", which is directly contradicted by the Gaussian reward assumption. The contribution states: "This correlation breaks the independence assumption commonly used in canonical BAI analyses, rendering existing algorithms and theoretical guarantees inapplicable". Again, this depends on the BAI algorithm under consideration. Obviously, considering track and stop is unfair; there is extensive study on causal bandits which exploit the causal structure for BAI. Yet another example: "Compared to this line of work, our correlated sampling approach is fundamentally different: it does not rely on any structural assumptions about the unknown reward function. This generality makes our method applicable to a broader range of problem settings" comes without any justification about what these broader applications are. Given the setting, I don't think that the correlation structure subsumes linear bandits, and hence, the term "broader" is misleading.

**Questions:**

1. After (73) the authors state results for $\rho\in(0,1)$. Are we not considering negative correlation? As far as I understand, $\rho>-\frac{1}{K-1}$ satisfies the positive semi-definiteness of the covariance matrix.

2. Can the authors explain why the strict inequality in (7) should hold? Can the terms $-\frac{2\rho}{\omega_a}$ and $\rho^2(\frac{1}{\omega_a} - \frac{1}{\omega_1})$ not cancel?

3. Immediately before (36), the authors mention that "it is straightforward to verify that $\nu^*=0$". Can the authors explain this step?

4. "The results demonstrate that CORSA outperforms Track-and-Stop in both probability of correct identification": a plot of probability of error versus number of samples is not very meaningful in the fixed confidence setting.

5. Comparing figures 1(b) and 1(d), convergence in sampling allocation is observed at least after 25,000 samples, whereas the average sample complexity in Figure 1(b) is reported around and below 1000 samples, which does not depict the claimed theoretical benefits fairly (as the sampling fractions have most likely not converged for the specific $\delta$).

---

### Official Review · Reviewer_Fe9x · 2025-10-28

**Soundness:** 3
**Presentation:** 3
**Contribution:** 3
**Rating:** 6
**Confidence:** 3

**Summary:**

This paper makes impactful theoretical innovations to the BAI field by introducing relevant sampling, with rigorous analysis and practical potential. However, it faces non-trivial limitations in computational efficiency, applicability, and experimental comprehensiveness.

**Strengths:**

The K-dimensional non-convex min-max problem is cleverly reduced to univariate optimization via interval decomposition, enabling efficient solving. The CORSA algorithm features a clear structure, adaptive sampling ratio tracking, and a correlation-aware stopping rule.

**Weaknesses:**

The homogeneous correlation assumption (identical coefficients across arms) is rarely satisfied in practice, with no concrete methods for heterogeneous correlation extension. Dependence on Common Random Numbers excludes certain online learning scenarios. The Gaussian reward (known variance) assumption limits applicability to non-Gaussian/unknown variance, discrete, or heavy-tailed rewards—extensions are not adequately addressed. Experiments are limited to three-armed problems and single queuing systems, lacking validation on large-scale tasks or complex environments. No comparisons with structure-exploiting methods (e.g., linear bandits) are provided, and the conditions for relevant sampling’s superiority over structural methods remain unexamined.

**Questions:**

Include comparisons with structure-exploiting methods across different problem settings, and clarify the conditions under which relevant sampling is preferred.

Add intuitive explanations (e.g., qualitative analysis of correlation impact) and visualizations (e.g., algorithm performance curves, correlation effect diagrams).

---

### Official Review · Reviewer_eY4d · 2025-10-31

**Soundness:** 2
**Presentation:** 2
**Contribution:** 2
**Rating:** 4
**Confidence:** 3

**Summary:**

The authors study best-arm identification in environments where the correlation is known, derive the corresponding lower bound, and propose a tracking algorithm named **CORSA**. Unlike the previous approach by Garivier and Kaufmann, which assumes no correlation, their method involves a non-convex optimization, for which they also describe the computational procedure in detail.

**Strengths:**

**Strengths:**
The paper studies best-arm identification (BAI) in environments where the correlation between arms is known and mathematically proves that when such correlations are available, it is possible to improve the ratio-type lower bound. The authors carefully describe how this ratio can be computed and rigorously show that the seemingly complex non-convex optimization can be reduced to (relatively) simple computations over $K$ intervals. The expressions for $w_a*$ and $w_1*$ presented in Theorem 2, which arise from the non-convex optimization process, appear quite interesting.

**Weaknesses:**

**Weaknesses:**

1. The assumptions are extremely restrictive. According to the paper, the learning agent is assumed to know the correlation coefficient $\rho$, which is not a tunable hyperparameter. Moreover, the paper claims that the results hold when the same correlation coefficient $\rho$ applies to *all* pairs of arms and that this setting can be easily extended, but there is no appendix or supplementary guidance supporting this claim.
2. In addition, the authors’ abstract is highly overstated. They explicitly claim that their method “requires no assumptions on the reward function or the arm structure,” which is simply false. As mentioned above, their results crucially rely on the strong assumption that the arms’ correlations are fixed at $\rho$ at every timestep *and* that this $\rho$ is known to the user. The authors also suggest that their method “always” outperforms Garivier and Kaufmann’s results, but this superiority arises entirely from exploiting the known correlation coefficient $\rho$. This critical assumption is never mentioned in the abstract, which is misleading. Until I read the main text carefully, I even thought they had developed some instance transformation technique allowing the user to choose $\rho$ freely. This omission feels deceptive—such a strong assumption must be stated clearly.
3. The notation is also questionable in terms of soundness. What does $x \in \mathcal{F}(x)$ even mean? The set itself is a function of $x$, so how can $x$ belong to it? Looking further only led to more confusion. In Theorem 2, the sets $\mathcal{K}_1$ and $\mathcal{K}_2$ depend on $x$, and $\mathcal{F}(x)$ is defined as the set of $x$’s that depend on $\mathcal{K}_2$. What kind of mathematical statement is this supposed to be?

There are a few additional points worth noting:

1. What is the relation to the 2021 paper *“Best-Arm Identification in Correlated Multi-Armed Bandits”*, which almost shares the same title? That paper immediately appears when you search this title online and seems highly relevant, yet there is not even a single citation or mention of it.
2. (Minor) The expression used to define $x*$ could be improved. In Eq. (11), the authors write “$\min g(x) = ...$”, but it is unclear whether this is defining $g(x)$ or the value of $\min g(x)$. A clearer formulation would be something like $x^* = \arg\min g(x) \text{ where } g(x) := ...$. Such notational issues raise concerns about the overall mathematical clarity of the paper.

**Overall:**
While the paper may contain a potentially meaningful contribution, the lack of adequate clarification on its restrictive assumptions and overstated claims makes it difficult to recommend acceptance. My evaluation therefore leans toward rejection.

**Questions:**

Please check the weakness above. Main issues I want to check are:

1) Assumption on $\rho$ + explanation about the overstatement in abstract
2) Mathematical descriptions
3) Comparison with Best-Arm Identification in Correlated Multi-Armed Bandits: Are the authors certain that their results are completely unrelated to this prior work? They claim to consider a “similarity structure” and even discuss various linear and generalized linear settings, yet they make no mention of that paper at all. This omission raises concerns about the thoroughness of the authors’ reference check.

---

### Note · Authors · 2025-11-23

I have read and agree with the venue's withdrawal policy on behalf of myself and my co-authors.